# Hierarchical protein backbone generation with latent and structure diffusion

## Abstract

We propose a hierarchical protein backbone generative model that separates coarse and fine-grained details. Our approach called LSD consists of two stages: sampling latents which are decoded into a contact map then sampling atomic coordinates conditioned on the contact map. LSD allows new ways to control protein generation towards desirable properties while scaling to large datasets. In particular, the AlphaFold DataBase (AFDB) is appealing due as its diverse structure topologies but suffers from poor designability. We train LSD on AFDB and show latent diffusion guidance towards AlphaFold2 Predicted Alignment Error and long range contacts can explicitly balance designability, diversity, and noveltys in the generated samples. Our results are competitive with structure diffusion models and outperforms prior latent diffusion models.

## 1 Introduction

A challenge across diffusion models for protein backbone generation has been scaling to large datasets: ideally benefiting from improved diversity and generalization, but this empirically results in unwanted biases from low quality protein structures (Huguet et al., 2024). In this work, we aim to develop a diffusion model that scales to the AlphaFold DataBase (AFDB) (Varadi et al., 2022) with the ability to control for desired properties. Previous approaches use *structure*-based diffusion models (SDMs) over atomic coordinates, but this presents challenges in respecting equivariance and physical constraints such as bond lengths and angles. Unfortunately, this can hinder optimization and generalization of deep learning models – recent works in structure prediction (Abramson et al., 2024), conformer generation (Wang et al.), and material design (Yang et al., 2023) have found improved results by removing equivariance and physical constraints.

We hypothesize SDMs can be improved by conditioning the generation process on sampled contact maps, defined as a 2D binary matrix representing whether each pair of residues are within a short distance of each other. A contact map is sufficient to describe a protein's fold topology while the coordinates can capture biomolecular conformations and elucidate a protein's function. Prior works have shown generative models fail to learn meaningful representations and can benefit from alignment (Yu et al., 2024) or high-level conditioning (Li et al., 2024). We follow the latter intuition to propose using latent diffusion models (LDMs) to generate contact maps followed by SDMs to generate atomic coordinates conditioned on the contact map (Fig. 1). Together, our method called LSD (**L**atent and **S**tructure **D**iffusion) allows for training over large datasets then guiding for desired properties.

Our main technical contribution is developing LDMs for contact map generation. We encode protein structures into a $L \times K$ dimensional latent space where $L$ is the protein length and $K$ is the latent dimensionality. The decoder then reconstructs the contact map given the latents. Next, we train a modified Diffusion Transformer (DiT) (Peebles & Xie, 2023a) to sample from learned latent space while FrameFlow (Yim et al., 2024a) is trained to generate atomic coordinates conditioned on contact maps and latents. Lastly, we demonstrate high-level control over the generated structures by guiding latent generation towards lower mean Predicted Alignment Error (PAE) and increased Long Range Contacts (LRC) for improved designability and diversity.

We evaluate LSD on protein backbone generation by training on the AlphaFold Database (AFDB) (Varadi et al., 2022) which we show is challenging to learn with only FrameFlow. Our results show LSD improves generalization to more diverse fold topologies which suggests combining LDMs and SDMs can be beneficial for scaling to large datasets like AFDB. We achieve improved results when

Figure 1: **Overview.** We propose separating protein structure generation into two stages. In the first stage, we generate coarse representations of proteins as a contact map using latent diffusion. High-level properties can be enforced in the contact map generation using guidance. In the second stage, we perform structure diffusion to generate structures conditioned on the coarse protein representation.

LSD is used with PAE or LRC guidance – demonstrating a novel capability of high-level protein generation control with LDMs. LSD is competitive with SDMs on AFDB and outperforms existing LDMs for protein backbone generation. Our contributions are summarized as follows:

1. We develop LSD, a novel hierarchical protein generative model that uses LDMs for contact map generation and SDMs for atomic coordinate generation.

2. We demonstrate the first instance of high-level guidance for protein backbone generation. PAE guidance improves designability while LRC guidance improves diversity and novelty.

3. When trained on AFDB, LSD is competitive with state-of-the-art SDMs on AFDB and outperforms the only publicly available LDMs for protein backbone generation.

## 2 BACKGROUND: LATENT DIFFUSION MODELS

Latent Diffusion Models (LDMs) use two components: an autoencoder to embed the data in a latent space and a diffusion model to generate samples from the latent space which are decoded back to data with the decoder (Vahdat et al., 2021; Rombach et al., 2022). In this section, we provide background of both components.

**Notation.** Superscript with parentheses denote time while subscripts are used to denote the index of a matrix or vector, i.e. $\mathbf{x}_i^{(t)}$ is the data at index $i$ and time $t$ in the diffusion process. Scalars and probability distributions will use subscripts for time.

### 2.1 AUTOENCODER: MAPPING DATA TO LATENT SPACE

Autoencoders consist of an encoder $p_\phi(\mathbf{z}|\mathbf{x})$ that maps data $\mathbf{x} \sim p_{\text{data}}$ into a latent variable $\mathbf{z}$ while the decoder $p_\psi(\mathbf{x}|\mathbf{z})$ maps $\mathbf{z}$ back to $\mathbf{x}$. The encoder and decoder are parameterized by neural networks with weights $\phi$ and $\psi$ respectively. Following (Vahdat et al., 2021), we use a Variational AutoEncoder (VAE) (Kingma, 2013) which is trained by minimizing the variational upper bound

$$\mathbb{E}_{p_{\text{data}}(\mathbf{x})}\left[\underbrace{\mathbb{E}_{p_\phi(\mathbf{z}|\mathbf{x})}\left[-\log p_\psi(\mathbf{x}|\mathbf{z})\right]}_{\text{Reconstruction}} + \underbrace{\lambda\mathbb{KL}\left[p_\phi(\mathbf{z}|\mathbf{x})||\mathcal{N}(0,I)\right]}_{\text{Regularization}}\right] \quad (1)$$

where $\lambda$ is the regularization weight for the Kullback-Leibler divergence ($\mathbb{KL}$). Rombach et al. (2022) utilized additional regularization terms. We plan to explore more regularizations in future work.

### 2.2 DIFFUSION MODEL OVER LATENT SPACE

Once the autoencoder is trained, we define $p^{(0)} = p_\phi$ as the target distribution of the latent diffusion process. Next, a diffusion model is trained to generate data by learning to map samples from Gaussian noise $\mathbf{z}^{(1)} \sim \mathcal{N}(0,I)$ with identity $I$ towards latents $\mathbf{z}^{(0)} \sim p_0$ which is then decoded back into data

$p_\psi(\mathbf{x}|\mathbf{z}^{(0)})$. The time variable $t \in [0,1]$ controls the mapping between noise and latents based on the time-dependent process[1]:

$$\mathbf{z}^{(t)} = \alpha(t)\mathbf{z}^{(0)} + \sigma(t)\mathbf{z}^{(1)}. \tag{2}$$

While many choices for $\alpha(t)$ and $\sigma(t)$ have been proposed, we use a simple linear interpolation popularized in flow models (Lipman et al., 2022; Liu et al., 2022; Albergo et al., 2023): $\alpha(t) = 1 - t$, $\sigma(t) = t$. The conditional distribution and score of eq. (2) can be analytically computed as

$$\nabla_{\mathbf{z}^{(t)}} \log q_t(\mathbf{z}^{(t)}|\mathbf{z}^{(0)}) = \frac{\mathbf{z}^{(t)} - \alpha(t)\mathbf{z}^{(0)}}{\sigma(t)^2} \quad \text{where} \quad q_t(\mathbf{z}^{(t)}|\mathbf{z}^{(0)}) = \mathcal{N}(\mathbf{z}^{(t)}; \alpha(t)\mathbf{z}^{(0)}, \sigma(t)^2 I). \tag{3}$$

We require $q_{t=1}(\cdot|\mathbf{z}^{(0)}) = p_1(\cdot)$ and $q_{t=0}(\cdot|\mathbf{z}^{(0)}) = p_0(\cdot)$. This allows learning to approximate the marginal score $\nabla_{\mathbf{z}^{(t)}} \log p_t(\mathbf{z}^{(t)})$ through the score matching objective (Hyvärinen & Dayan, 2005). Equivalently, we optimize the denoising autoencoder objective (Vincent, 2011)

$$\mathbb{E}_{p_0}[\mathbf{z}^{(0)}|\mathbf{z}^{(t)}] \approx \underset{\hat{\mathbf{z}}_\theta}{\arg\min} \, \mathbb{E}_{\substack{q_t(\mathbf{z}^{(t)}|\mathbf{z}^{(0)}) \\ \mathcal{U}(t;0,1) \\ p_0(\mathbf{z}^{(0)})}} \left[ \frac{1}{\sigma(t)^2} \|\hat{\mathbf{z}}_\theta(\mathbf{z}^{(t)}, t) - \mathbf{z}^{(0)}\|_2^2 \right]$$

where $\mathcal{U}$ is the uniform distribution and $1/\sigma(t)^2$ is a weighting term to encourage equal loss weighting across $t$. $\hat{\mathbf{z}}_\theta$ is a neural network with weights $\theta$ trained to predict the true latents. Using eq. (3), the marginal score can then be approximated as $s_\theta(\mathbf{z}^{(t)}, t) = \frac{\mathbf{z}^{(t)} - \alpha(t)\hat{\mathbf{z}}_\theta(\mathbf{z}^{(t)}, t)}{\sigma(t)^2}$. Once the score is learned, we can obtain samples $\mathbf{z}^{(0)}$ by integrating the reverse SDE starting with $\mathbf{z}^{(1)}$ towards $\mathbf{z}^{(0)}$,

$$d\mathbf{z}^{(t)} = \left[ a(t)\mathbf{z}^{(t)} - b(t)^2 s(\mathbf{z}^{(t)}; t) \right] dt + \gamma \cdot b(t) d\mathbf{w}^{(t)} \tag{4}$$

where $\mathbf{w}^{(t)}$ is a Wiener process, $a(t) = \frac{\partial}{\partial t} \log \alpha(t)$, and $b(t) = 2\sigma(t)(\frac{\partial \sigma(t)}{\partial t} - a(t)\sigma(t))$. $\gamma$ is a scale to control the variance of the noise which by default is set to $\gamma = 1$. Prior works have found setting $\gamma < 1$ to improve sample quality (Ajay et al., 2022; Yim et al., 2023). In this work, we use Euler-Maruyama integrator for all samples. We provide a derivation in App. A that eq. (4) is the reverse SDE of the interpolant in eq. (2).

## 3 METHOD: LATENT AND STRUCTURE DIFFUSION (LSD)

In this section, we present our method for hierarchical protein backbone generation. The method consists of three components: a structure-to-contact autoencoder (Sec. 3.1), LDM to sample latents from the autoencoder latent space and a SDM to samples structures from the sampled latents (Sec. 3.2). We discuss training and sampling in Sec. 3.3. Lastly, we discuss PAE and LRC guidance in Sec. 3.4.

### 3.1 STRUCTURE-TO-CONTACT AUTOENCODER

We denote a protein backbone's atomic coordinates as $\mathbf{x} \in \mathbb{R}^{L \times 3 \times 3}$ where $L$ is the length of the protein (number of residues), 3 corresponds to the Nitrogen, Carbon$_\alpha$, and Carbon atom in each residue. For our encoder, we use the ProteinMPNN (Dauparas et al., 2022) architecture to embed $\mathbf{x}$ into a latent $p_\phi(\mathbf{x}) = \mathbf{z} \in \mathbb{R}^{L \times K}$ where $K$ is the latent dimensionality with $K \ll L$. We modify ProteinMPNN to output the mean and variance Gaussian parameters of each latent.

We aim to learn a coarse representation of $\mathbf{x}$ in the latent space. Our approach is to train the decoder to predict the contact map $\mathbf{c} \in \{0,1\}^{L \times L}$ where $\mathbf{c}_{ij} = 1$ if the distance between the Carbon$_\alpha$ atoms of $\mathbf{x}_i, \mathbf{x}_j$ is less than 8Å and $\mathbf{c}_{ij} = 0$ otherwise.[2] The decoder $p_\psi$ takes the Kronecker product of the latents and predicts contact map probabilities: $p_\psi(\mathbf{z}_i \otimes \mathbf{z}_j)$ for all $i, j$. We parameterize $p_\psi$ with a 3 layer multi-layer perceptron with ReLU activations. The Kronecker product $\mathbf{z}_i \otimes \mathbf{z}_j \in \mathbb{R}^{K \times K}$ is the matrix of all possible products between the entries of $\mathbf{z}_i$ and $\mathbf{z}_j$. The encoder and decoder are trained

---

[1]Technically each $\mathbf{z}^{(t)}$ qualifies as a latent variable. However, we will strictly refer to $\mathbf{z}^{(0)}$ as latents in our context since these directly map to data via the decoder.

[2]8Å is commonly used to define a contact (Hopf et al., 2014)

using eq. (1) where the reconstruction term is modified to minimize the negative log-likelihood of the ground truth contacts:

$$\mathbb{E}_{p_\phi(\mathbf{z}|\mathbf{x})}\Big[\frac{1}{|\mathcal{Z}_0|}\sum_{(i,j)\in\mathcal{Z}_0} -\log p_\psi(\mathbf{c}_{ij}=0|\mathbf{z}_i\otimes\mathbf{z}_j) + \frac{1}{|\mathcal{Z}_1|}\sum_{(i,j)\in\mathcal{Z}_1} -\log p_\psi(\mathbf{c}_{ij}=1|\mathbf{z}_i\otimes\mathbf{z}_j)\Big]$$

where $\mathcal{Z}_0 = \{(i,j) : \mathbf{c}_{ij} = 0\}$ and $\mathcal{Z}_1 = \{(i,j) : \mathbf{c}_{ij} = 1\}$ are the set of indices where $\mathbf{c}_{ij} = 0$ and $\mathbf{c}_{ij} = 1$ respectively. Due to the sparsity of the contacts, we weight each class separately by its propensity. For short-hand notation, we will refer to the full decoded contact probabilities as $\hat{\mathbf{c}}_\psi(\mathbf{z}) \in [0, 1]^{L\times L}$ where $\hat{\mathbf{c}}_\psi(\mathbf{z})_{ij} = p_\psi(\mathbf{z}_i\otimes\mathbf{z}_j)$.

## 3.2 LATENT AND STRUCTURE DIFFUSION MODELS

The LDM requires learning the latent score function $s_\theta(\mathbf{z}^{(t)}, t)$ where $\mathbf{z}^{(t)}$ is a noisy version of the encoded latents $\mathbf{z}^{(0)} = p_\phi(\mathbf{x})$ as defined in eq. (2). For neural network architecture, we use the Diffusion Transformer (DiT) (Peebles & Xie, 2023b) since it is successfully used across computer vision. We adapt DiT for our purposes by treating each residue latent $\mathbf{z}_i^{(t)}$ as a token. We use Rotary Positional Encodings (RoPE) (Su et al., 2024) instead of absolute positional encodings as done in Hayes et al. (2024).

The SDM is a modified version of FrameFlow (Yim et al., 2024a) that is trained to predict the denoised atomic coordinates $\hat{\mathbf{x}}_\varphi(\mathbf{x}^{(t)}, \hat{\mathbf{c}}_\psi(\mathbf{z}^{(0)}), \mathbf{z}^{(0)}, t)$ where $\varphi$ are the FrameFlow neural network weights. We condition the SDM by concatenating $\mathbf{z}^{(0)}$ and $\hat{\mathbf{c}}_\psi(\mathbf{z}^{(0)})$ to the initial set 1D and 2D features provided to FrameFlow. FrameFlow uses flow matching over SE(3) (Chen & Lipman, 2024) which is equivalent to the probabilistic Ordinary Differential Equation (ODE) perspective of diffusion models. We follow the training and sampling procedure of FrameFlow with the addition of our latent conditioning, no self-conditioning, and no rotation annealing. We choose FrameFlow over other SDMs due to the ease of its open sourced code but any other SDM can be used in our framework. Due to the architecture choice, our LDM is SE(3) invariant while the SDM is SE(3) equivariant.

## 3.3 MULTI-STAGE TRAINING AND SAMPLING

The VAE training loss is described in Sec. 3.1 while the LDM and SDM losses are described in Sec. 3.2. While end-to-end training of all models is possible, we found this to be unstable and difficult to optimize. We instead use a training procedure inspired by Rombach et al. (2022) where the autoencoder is frozen during latent diffusion training. It involves three stages: (1) pre-training the VAE by itself, (2) jointly training the VAE and SDM, and (3) freezing the VAE weights and training only the LDM. In our experiments, we use the same optimizer and learning rate across all stages. App. B.1 provides more details on the training setup.

To sample, we first generate latents $\mathbf{z}^{(0)}$ from the LDM using eq. (4) and obtain the contact map $\hat{\mathbf{c}}_\psi(\mathbf{z}^{(0)})$. Both $\mathbf{z}^{(0)}$ and $\hat{\mathbf{c}}_\psi(\mathbf{z}^{(0)})$ are then provided to the SDM to sample atomic coordinates conditioned on $\hat{\mathbf{c}}_\psi(\mathbf{z}^{(0)})$ using the SE(3) flow, see Yim et al. (2024a). Fig. 1 illustrates the sampling process. In the next section, we describe sampling with guidance towards desired high-level properties.

## 3.4 PAE AND LRC GUIDANCE

Let $\mathbf{y}$ denote a property such as a class label associated with each latent $\mathbf{z}^{(0)}$. Dhariwal & Nichol (2021) proposed to train a classifier to predict the property from each noised latent $\mathbf{z}^{(t)}$ which is then used to guide the LDM towards a desired class label. This is achieved using Bayes rule to approximate the property conditioned score,

$$\nabla_{\mathbf{z}^{(t)}}\log p_t(\mathbf{z}^{(t)}|\mathbf{y}) = \nabla_{\mathbf{z}^{(0)}}\log p_t(\mathbf{z}^{(t)}) + \nabla_{\mathbf{z}^{(t)}}\log p_t(\mathbf{y}|\mathbf{z}^{(t)}) \approx s_\theta(\mathbf{z}^{(t)};t) + s(\mathbf{z}^{(t)};t) \quad (5)$$

where $s(\mathbf{z}^{(t)};t)$ is parameterized to approximate $\nabla_{\mathbf{z}^{(0)}}\log p_0(\mathbf{y}|\mathbf{z}^{(t)})$. We then substitute eq. (5) as the score into eq. (8) to approximately sample from $p(\mathbf{z}^{(0)}|\mathbf{y})$,

$$d\mathbf{z}^{(t)} = \Big[a(t)\mathbf{z}^{(t)} - b(t)^2\Big(s_\theta(\mathbf{z}^{(t)};t) + s(\mathbf{z}^{(t)};t)\Big)\Big]dt + \gamma\cdot b(t)d\mathbf{w}^{(t)}. \quad (6)$$

We describe multiple options of $s(\mathbf{z}^{(t)};t)$ for guiding towards PAE and long range contacts.

**Long range contact (LRC).**   Protein generative models often exhibit a preference for predominantly alpha-helical structures – due to the prevalence of alpha-helices in protein datasets (Dawson et al., 2017) – which can limit the diversity of generated fold topologies. To address this bias, we use guidance towards more LRCs by leveraging the decoder's contact map predictions $p_\psi$. Following Hayes et al. (2024), a LRC is defined as a contact $\mathbf{c}_{ij}$ with sequence distance greater than 12. Let $\mathcal{Z}_{\text{LR}} = \{(i,j) : |i - j| > 12\}$ be the set of pairwise indices with sequence distance greater than 12. With $\mathbf{y} = \{\mathbf{c}_{ij} = 1 \; \forall (i,j) \in \mathcal{Z}_{\text{LR}}\}$ as the LRC property, we define the LRC guidance score

$$s_{\text{LRC}}(\mathbf{z}^{(t)}; t) = e^{-r_{\text{LRC}} \cdot (1-t)} \cdot \nabla_{\mathbf{z}^{(t)}} \left[ \frac{1}{|\mathcal{Z}_{\text{LR}}|} \sum_{(i,j) \in \mathcal{Z}_{\text{LR}}} \log p_\psi \left( \mathbf{c}_{ij} = 1 | \hat{\mathbf{z}}_\theta(\mathbf{z}^{(t)}, t)_i \otimes \hat{\mathbf{z}}_\theta(\mathbf{z}^{(t)}, t)_j \right) \right]$$

where $r_{\text{LRC}} \in \mathbb{R}$ is a hyperparameter controlling the decay of the score coefficient. To perform guidance towards more LRCs, we substitute $s_{\text{LRC}}(\mathbf{y}, \mathbf{z}^{(t)}; t)$ for $s(\mathbf{z}^{(t)}; t)$ in eq. (6).

**Predicted Alignment Error (PAE).**   Since lower PAE is often correlated with protein design success (Bennett et al., 2023), we are interested in generating protein structures with lower PAE. Our main metric, designability, also correlates with lower PAE (see Fig. 10). We train a neural network $f_\vartheta(\hat{\mathbf{z}}_\theta(\mathbf{z}^{(t)}, t), t)$ with weights $\vartheta$ to predict the global average PAE of the structure $\mathbf{x}$ corresponding to each latent noised latent $\mathbf{z}^{(t)}$. See App. B.2 for data curation, architecture, and training details. Since PAE is a scalar value, classifier guidance does not directly work. Instead, we substitute $p_t(\mathbf{y}|\mathbf{z}^{(t)})$ with a Boltzmann distribution that assign high probability to lower PAE predictions: $p_t^{\text{PAE}}(\mathbf{z}^{(t)}) \propto e^{-\omega_{\text{PAE}} \cdot f_\vartheta(\hat{\mathbf{z}}_\theta(\mathbf{z}^{(t)}, t), t)}$ with weight $\omega_{\text{PAE}} \in \mathbb{R}$. We define the PAE gudiance score as

$$s_{\text{PAE}}(\mathbf{z}^{(t)}, t) = \nabla_{\mathbf{z}^{(t)}} \log p_t^{\text{PAE}}(\mathbf{z}^{(t)}) = -\omega_{\text{PAE}} \nabla_{\mathbf{z}^{(t)}} f_\vartheta(\hat{\mathbf{z}}_\theta(\mathbf{z}^{(t)}, t).$$

While not principled, we find $s_{\text{PAE}}$ intuitive in guiding towards lower PAE and works well in practice. To perform guidance towards lower PAE, we substitute $s_{\text{PAE}}$ for $s$ in eq. (6) and sweep over different $\omega_{\text{PAE}}$ values in the experiments.

**Joint guidance.**   Lowering PAE is correlated with improved designability but could come at the cost of diversity and novelty. On the other hand, increasing LRCs results in more diverse, novel protein structures but could hurt designability. Thus it makes sense to combine both. We will make use of the following score when guiding towards both properties,

$$s_{\text{J}}(\mathbf{z}^{(t)}; t) = s_{\text{PAE}}(\mathbf{z}^{(t)}; t) + s_{\text{LRC}}(\mathbf{z}^{(t)}; t)$$

where we still have two hyperparameters $\omega_{\text{PAE}}$ and $r_{\text{LRC}}$ to control the strength of each guidance.

## 4   RELATED WORK

**Latent Diffusion Models (LDMs).**   Latent Score-based Generative Model (LSGM) (Vahdat et al., 2021) first proposed using a combination of a VAE and diffusion model over the latent space. Stable diffusion (Rombach et al., 2022) extended LSGM with architectural and training improvements that achieved state-of-the-art (SOTA) results in image synthesis. DiT (Peebles & Xie, 2023a) and SiT (Ma et al., 2024) further improved the scalability with a Transformer (Vaswani, 2017) architecture tailored for diffusion models.

The success of LDMs has motivated their use in protein applications. OmniProt (McPartlon et al., 2024) is a LDM for protein-protein docking but with no open source code. LatentDiff (Fu et al., 2023) is the only other LDM for protein structure generation to the best of our knowledge. We also consider methods FoldToken (Gao et al., 2024) and ESM3 (Hayes et al., 2024) that learn discrete latent tokens and use autoregressive masked language models to be discrete LDMs for protein structure generation. Likewise, DiffTopo (Correia, 2024) and TopoDiff (Zhang et al., 2023) do not exactly fit the LDM framework but are related by using diffusion to sample a coarse protein fold topology followed by a diffusion model to produce a structure conditioned on the topology. DiffTopo and TopoDiff do not have open source code to compare with. In Sec. 5, we use LatentDiff and ESM3 as baselines.

**Structure Diffusion Models (SDMs).**   RFdiffusion (Watson et al., 2022) is a widely used SDM with proven results in real-world protein design applications. Numerous other SDMs have been

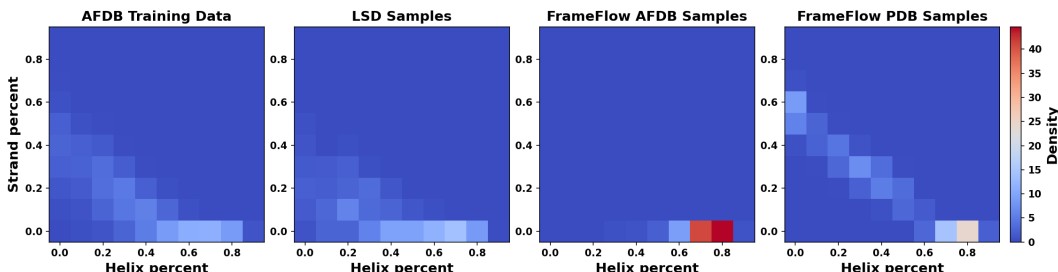

Figure 2: Secondary structure distribution of the training data compared to samples from LSD Frameflow trained on AFDB or PDB. 10 samples of each length between 60-128 were generated with 100timesteps each. LSD used $\gamma = 1$ while no modifications were made for FrameFlow. We computed helix and strand percents of each sample then produced a 2D histogram distribution with 10 bins along each axis. See Sec. 5.2 for discussion.

developed such as GENIE2 (Lin et al., 2024), FoldFlow2 (Huguet et al., 2024), Chroma (Ingraham et al., 2023), MultiFlow (Campbell et al., 2024), and AlphaFold3 (Abramson et al., 2024). See Yim et al. (2024b) for a survey of SDMs. As mentioned in Sec. 1, a challenge with SDMs has been scaling to large datasets while mitigating unwanted biases from low quality protein structures. We show LSD is a novel approach to train on a large dataset, AFDB, with varying data quality and use guidance to control protein properties. Since our approach is built on top of FrameFlow, we show in Sec. 5 that LSD improves upon FrameFlow's limitations when training on AFDB. We include RFdiffusion and GENIE2 as reference points of SOTA protein structure generation methods. Since MultiFlow is a co-design extension of FrameFlow, we use FrameFlow's results as representative of MultiFlow's performance. Lastly, we benchmark against ProteinSGM (Lee et al., 2023), a diffusion model over pairwise distances and dihedral angles.

## 5 EXPERIMENTS

In this section, we run experiments with LSD to analyze its performance on protein structure generation. Sec. 5.1 describe our training and evaluation set-up. Sec. 5.2 analyzes LSD with ablations and demonstrates improved results over only using FrameFlow. Sec. 5.3 then demonstrates capabilites with PAE and contact guidance to control high-level properties. Lastly, Sec. 5.4 compares LSD to prior protein structure generation baselines discussed in Sec. 4.

### 5.1 SET-UP

**Training Details.** We train LSD on the Foldseek (Van Kempen et al., 2024) clustered AlphaFold DataBase (AFDB) (Varadi et al., 2022) as done in GENIE2 (Lin et al., 2024). We filter out examples that are longer than 128 residues and minimum pLDDT (AlphaFold2 predicted confidence metric) lower than 80. The latter is a commonly used filter to remove low quality protein structures from AFDB but is not suffucient (Varadi et al., 2022). This results in 282936 training examples. Dimensions and training details of all our neural networks are provided in App. B.1.

**Evaluation Details.** For each method, we sample 10 proteins of each length between 60-128. Standard metrics for protein backbone generation are designability (**Des**), diversity (**Div**), and novelty (**Nov**) as described in Yim et al. (2023). Novelty was computed against the AFDB database. Designable Pairwise TM-score (**DPT**) is defined as the average pairwise TM-score (Zhang & Skolnick, 2004) between designable proteins. We include **H/S/C** as the average helix, strand and coil secondary structure composition (Kabsch & Sander, 1983) of the designable samples, i.e. 60/10/20 means each structure is made up of 60% helix, 10% strands, and 20% coil on average. Designability is not a accurate indicator of how well a generative model matches the training distribution since the training dataset is far from 100% designable (Huguet et al., 2024). Instead, we measure how well a protein structure generative model captures the training distribution by computing Secondary Structure Distance (**SSD**), defined as the Wasserstein distance between the discretized secondary structure distribution of the training dataset and the generated proteins *with no designability filtering*. Alg. 1 describes how we compute SSD. See App. B.3 for more explanation of our metrics.

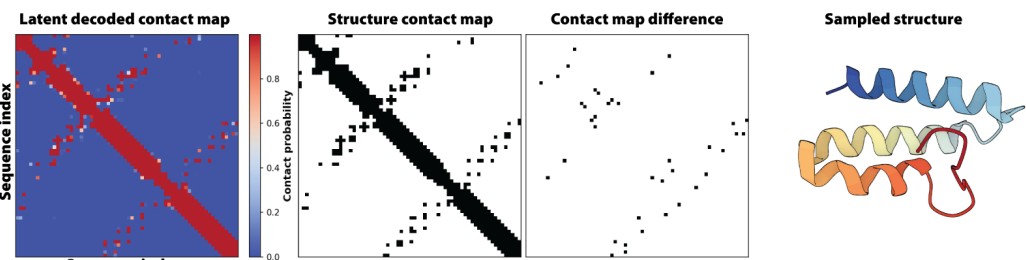

Figure 3: Visualization of the contact map after sampling latents and structures with LSD. We observe the agreement between the latent decoded contact map and the structure contact map is high. The PRAUC and ROCAUC is on average 0.99 and 0.92 between the latent decoded contact map and structure contact map in our evaluation benchmark with $\gamma$=1.

## 5.2 LSD ANALYSIS

**Hyperparameter and ablations.** We swept over the number of latents $K$ and $\mathbb{KL}$ regularization weight $\lambda$ in eq. (1) to select the best setting based on performance in the VAE pre-training stage. To evaluate, we held out 32 random protein clusters based on Foldseek and computed the decoder's ROCAUC and PRAUC of long range contacts (LRC) defined as all contacts $\mathbf{c}_{ij}$ with $|i - j| > 12$ which are visualized in Fig. 4. We found LRC performance to be most indicative of autoencoder performance. Our results are shown in tab. 5 where we find $K = 4$ and $\lambda = 0.1$ to be optimal. We next ablated architecture choices of the LDM by removing RoPE and using a standard Transformer instead of DiT. Tab. 6 shows RoPE and DiT all contribute to achieve the best performance. We sweep over noise scales in tab. 8 where we show $\gamma = 0.7$ gives the best designability and diversity trade-off.

**LSD samples diverse structures while FrameFlow AFDB collapses to alpha helices.** We consider two versions of FrameFlow: the published FrameFlow PDB trained on the Protein Data Bank (PDB)[3] (Berman et al., 2000) and FrameFlow AFDB where we re-trained FrameFlow on the same dataset as LSD. We then sampled both FrameFlows and LSD with the procedure described in Sec. 5.1 and focus on their secondary structure distributions. Fig. 2 shows that FrameFlow PDB samples a spread of helix and strand compositions while FrameFlow AFDB collapses to almost always sampling alpha helices despite the AFDB training data having a diverse secondary structure distribution. Prior works Huguet et al. (2024); Lin et al. (2024) have found neural network modifications such as triangle layers to improve performance on AFDB but this incurs cubic memory consumption. Instead, LSD uses a hierarchical approach to improve generalization to diverse fold topologies. These results indicate the contact map generation with LDM is beneficial to help induce diverse protein folds in the subsequent generation of protein structures with FrameFlow.

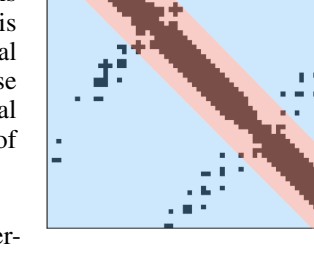

Figure 4: Visualization of long range contacts (LRC) in blue.

**Contact map and structure generation are consistent.** We verify the generated structures from the SDM are consistent with the conditioned contact maps from the LDM; in other words, we check the SDM is not ignoring the contact map. The LDM first samples latents $\mathbf{z}^{(0)}$ which are provided to the decoder to produce a *latent decoded contact map* $\hat{\mathbf{c}}_\psi(\mathbf{z}^{(0)})$. Conditioned on $\hat{\mathbf{c}}_\psi(\mathbf{z}^{(0)})$, the SDM samples structures $\mathbf{x}$ from which we can compute the binary *structure contact map* $\mathbf{c}(\mathbf{x}) \in \{0,1\}^{L \times L}$ and look at the *contact map difference* $|\mathbf{c}(\mathbf{x}) - \arg\max(\hat{\mathbf{c}}_\psi(\mathbf{z}^{(0)}))|$. Fig. 3 shows a visualization of these quantities. If SDM is ignoring the contact map, we would expect many contact map differences which is not the case visually. Quantititatively, the ROCAUC and PRAUC of $\mathbf{c}(\mathbf{x})$ and $\hat{\mathbf{c}}_\psi(\mathbf{z}^{(0)})$ are 0.99 and 0.92 respectively on average when sampling in the protein backbone generation benchmark. This indicates high consistency between the contact map and the generated structures.

---

[3]Weights were downloaded from https://github.com/microsoft/protein-frame-flow

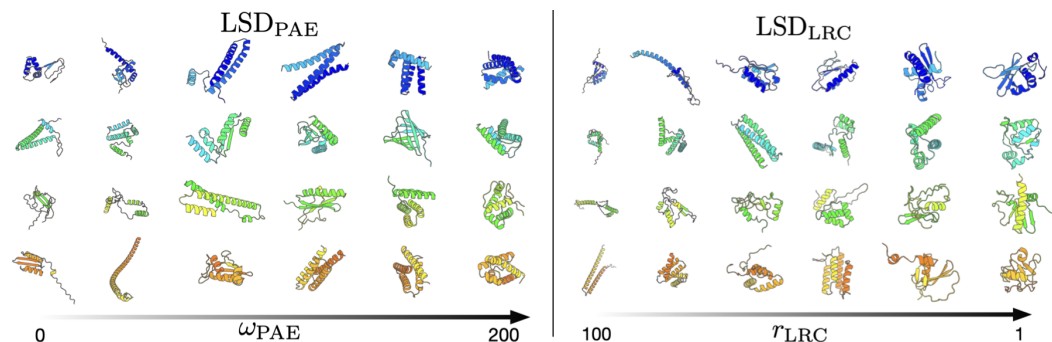

Figure 5: Samples at $\gamma$=1 with different guidance scales. **LSD$_{\text{PAE}}$**: Increasing $\omega_{\text{PAE}}$ leads to lower PAE of sampled structures with more secondary structure. **LSD$_{\text{LRC}}$**: Decreasing $r_{\text{LRC}}$ leads to more globular and diverse folds. We see $r_{\text{LRC}} = 1$ leads to increase of coils.

## 5.3 LSD GUIDANCE

We next explore the ability to control the properties of the generated samples using guidance. Here we use $\gamma = 1$ to study guidance under the correct reverse SDE eq. (8). In Tab. 1, we evaluate structure generation for different guidances and parameters.

Table 1: LSD results with $\gamma = 1$. We use 100 timesteps for both latent and structure diffusion.

| Method | $\omega_{\text{PAE}}$ | $r_{\text{LRC}}$ | Des ($\uparrow$) | Div ($\uparrow$) | DPT ($\downarrow$) | Nov ($\downarrow$) | SSD ($\downarrow$) | H/S/C |
|---|---|---|---|---|---|---|---|---|
| LSD | | | 15% | 84 | 0.6 | 0.74 | **0.13** | 61/10/28 |
| LSD$_{\text{PAE}}$ | 10 | | 31% | 133 | 0.54 | 0.7 | 0.25 | 61/12/28 |
| | 50 | | 66% | 204 | 0.48 | 0.68 | 0.58 | 67/8/25 |
| | 100 | | **76%** | 202 | 0.46 | 0.66 | 0.78 | 69/7/24 |
| LSD$_{\text{LRC}}$ | | 10 | 15% | 89 | 0.62 | 0.67 | 0.23 | 52/16/31 |
| | | 5 | 16% | 94 | 0.63 | 0.66 | 0.32 | 40/23/35 |
| | | 1 | 10% | 62 | 0.66 | **0.61** | 0.24 | 34/26/40 |
| LSD$_{\text{J}}$ | 50 | 5 | 61% | **217** | **0.51** | 0.65 | 0.22 | 56/21/30 |

Each variant shows a different property being optimized. Using no guidance (LSD) gives the best fit to the training distribution as indicated by the lowest SSD value. PAE guidance (LSD$_{\text{PAE}}$) shows designability increases as $\omega_{\text{PAE}}$ increases but structures become more helical. Fig. 11 demonstrates that increasing $\omega_{\text{PAE}}$ leads to decreasing mean PAE values across varying lengths as computed by AlphaFold2. LRC guidance (LSD$_{\text{LRC}}$) gives the best novelty and more strands as the weight decay rate $r_{\text{LRC}}$ decreases but suffers from low designability. We visualize the structures for PAE and LRC guidance in Fig. 5 as $\omega_{\text{PAE}}$ and $r_{\text{LRC}}$ vary. The structures reflect the **H/S/C** values where PAE guidance leads to more "simple" helical structures and LRC guidance leads to more diverse fold topologies. Using both guidances (LSD$_{\text{J}}$) reflects a balance between all metrics while achieving the best diversity. In summary, we are able to control for different properties using a single diffusion model and guidance techniques.

Analyzing the contact map diffusion trajectories $\hat{\mathbf{c}}_\psi(\hat{\mathbf{z}}_\theta(\mathbf{z}^{(t)}))$ across $t$ leads to insights into how guidance affects the generation process. Fig. 6 shows a prototypical trajectory for each variant. All trajectories start with a blurred contact map at $t = 1.0$ that sharpens as $t = 0.0$. We see PAE guidance encodes helices for most residues early on. PAE guided trajectories tend to encourage short range contacts (near the diagonal) at the beginning while LRC guided trajectories encourage long range contacts (far from the diagonal).

## 5.4 PROTEIN STRUCTURE GENERATION BENCHMARK

We benchmark our best settings against previous protein structure generative models for backbone generation. We compare against both LDMs and SDMs described in Sec. 4. However, LDMs are the

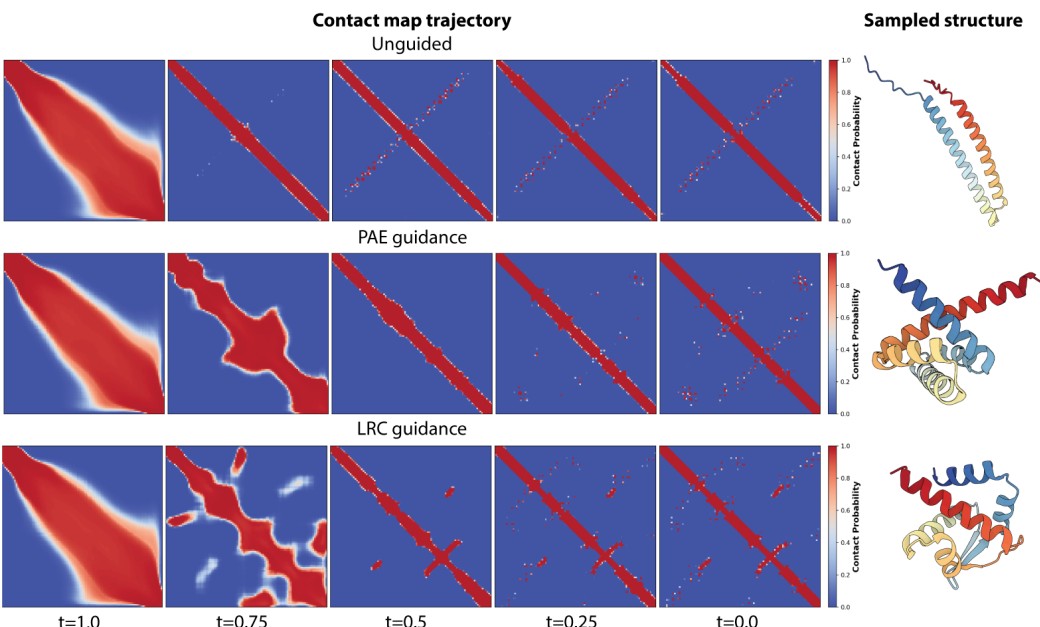

Figure 6: **Contact map diffusion trajectories.** We show prototypical examples of how the contact map evolves over time as latent diffusion progresses by visualizing the latent decoded contact map $\hat{\mathbf{c}}_\psi(\hat{\mathbf{z}}_\theta(\mathbf{z}^{(t)}))$ as $t$ goes from 1.0 to 0.0. Each row corresponds to a different guidance variant discussed in Sec. 5.3. At the far right, we show the structure generated from the final latent $\mathbf{z}^{(0)}$.

most direct comparison since contact map generation with LDM is the main step in LSD where most of the protein is determined. App. C.3 describes how we ran each baseline using their open source implementation. As discussed in Sec. 5.2, our best results are achieved with $\gamma = 0.7$. Analogous to Sec. 5.3, we ran a sweep to find the best hyperparameters at $\gamma = 0.7$ for LSD$_{\text{PAE}}$, LSD$_{\text{LRC}}$, and LSD$_{\text{J}}$; see App. C.3 for details and hyperparameters. We found more timesteps to not give improvements worth the extra compute. Tab. 2 shows our results.

Table 2: Protein backbone generation results. * LatentDiff does not allow for controlling the length of generated proteins since it sample the length. Out of 10,000 samples, we were unable to sample above length 100. Therefore, only 10 proteins per length 60-100 were evaluated for LatentDiff.

| Type | Method | Des (↑) | Div (↑) | DPT(↓) | Nov(↓) | SSD(↓) | H/S/C |
|------|--------|---------|---------|--------|--------|--------|-------|
| | RFdiffusion | 96% | 247 | 0.43 | 0.71 | 0.99 | 78/7/15 |
| | ProteinSGM | 49% | 122 | 0.37 | | 0.51 | 61 /10 /29 |
| SDM | FrameFlow PDB | 91% | 278 | 0.48 | 0.65 | 0.35 | 52/20/27 |
| | FrameFlow AFDB | 23% | 54 | 0.42 | 0.70 | 1.32 | 77 /0 /23 |
| | GENIE2 | **97%** | **369** | 0.51 | **0.62** | 0.84 | 47/7/24 |
| Lang. | ESM3 | 61% | 127 | **0.37** | 0.84 | **0.21** | 60/11/29 |
| LDM | LatentDiff* | 17% | 34 | 0.51 | 0.73 | 0.75 | 74/2/25 |
| | LSD ($\gamma$=0.7) | 69% | 203 | 0.46 | 0.74 | 0.86 | 76/4/20 |
| LDM+ | LSD$_{\text{PAE}}$ ($\gamma$=0.7) | **94%** | 204 | **0.42** | 0.71 | 1.03 | 75/4/20 |
| SDM | LSD$_{\text{LRC}}$ ($\gamma$=0.7) | 33% | 182 | 0.59 | **0.61** | **0.24** | 44/21/35 |
| | LSD$_{\text{J}}$ ($\gamma$=0.7) | 74% | **296** | 0.53 | 0.66 | 0.26 | 57/15/28 |

Our first observation is that all LSD variants outperform LatentDiff on all metrics thus achieves state-of-the-art (SOTA) performance for LDMs. LSD$_{\text{J}}$ beats ESM3 on all metrics except DPT and SSD. SDMs are known to be SOTA in protein backbone generation – we focus on comparing to GENIE2

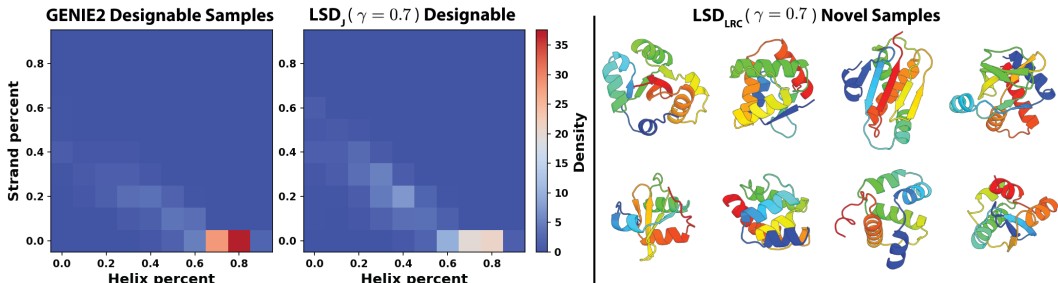

Figure 7: **Left**: Secondary structure distribution of designable samples from GENIE2 and LSD$_J$. Despite GENIE2's increased diversity, we see LSD$_J$ achieves more diverse folds in terms of secondary structure. **Right**: Novel samples with diverse folds from LSD$_{LRC}$ where "novel" is defined as designable and <0.5 max TM-score to the AFDB as computed by Foldseek. We show 8 out of 19 novel samples from LSD$_{LRC}$. Notice the diverse secondary structure topologies while staying novel.

which achieves the best overall results. GENIE2 is impressive in achieving SOTA performance in most categories with a single setting. LSD requires different guidances to be competitive in each category. Looking at H/S/C, we see LSD$_J$ achieves two times higher beta strand composition indicating more diverse secondary structure compositions than GENIE2 and adheres to the training closer indicated by the lower SSD. Fig. 7 shows the difference in designable secondary structure composition and shows the diverse fold topologies in novel samples from LSD$_{LRC}$ which achieves the best novelty score. We note GENIE2 is a far more expensive model that makes use of $O(L^3)$ memory intensive triangle update layers (Jumper et al., 2021). One length 100 protein generation takes **2.3 min.** for GENIE2 while LSD$_J$ takes **0.3 min.** on a Nvidia A6000 GPU. We found using triangle updates to substantially slow down training and research iteration. We plan to investigate different architectures and scaling up LSD.

LSD achieves a significant improvement over previous LDMs to close the performance gap between LDMs and state-of-the-art SDMs by combining the two. Compared to only FrameFlow, LSD allows options to guide generation towards which property we wish to optimize. More novelty? Use LSD$_{LRC}$. More designable? Use LSD$_{PAE}$. Higher diversity with diverse H/S/C? Use LSD$_J$. Our approach demonstrates versatility of controlling generation towards various properties. A natural extension is to consider properties such as binding affinity, thermostablity, or catalytic activity.

# 6 DISCUSSION

The goal of LSD is to develop a new framework for protein structure generation capable of separating high- and low-level details during the generation process. We combined latent and structure diffusion to break up the generative procedure into first sampling latents, contact maps, and finally the atomic coordinates. We showed how including intermediate contact maps helps learn large datasets such as AFDB and how guidance techniques can improve the quality of the generated structures. We compared LSD to existing protein structure generation methods and showed that it is competitive with state-of-the-art SDMs and outperforms prior LDMs for protein backbone generation.

Our limitations include not achieving state-of-the-art performance on all metrics with a single setting compared to SDMs and limiting LSD to unconditional backbone generation. We propose several directions to address these. First, performance can likely be improved with further investigation into the neural network architectures, i.e. triangle update layers that are widely utilized in Huguet et al. (2024); Lin et al. (2024) or swapping out FrameFlow with GENIE2 as the SDM in our framework. Second, we can optimize the LDM with the latest techniques report in the LDM for computer vision literature such as Autoguidance (Karras et al., 2024) and Stable Diffusion 3 (Esser et al., 2024). Lastly, we plan to extend our method to all-atomic biomolecular generation (Abramson et al., 2024) and design tasks – binder design and motif-scaffolding (Krishna et al., 2024). Extending to protein complexes involving multiple chains will require scaling up the model size to handle larger proteins. Our latent space only encode protein backbone coordinates but a natural extension is to include side-chain coordinates and protein sequence information. Having a malleable latent space opens up new possibilities for protein generative modeling and design.

## 7 REPRODUCIBILITY STATEMENT

Our implementations uses FrameFlow's open source code https://github.com/microsoft/protein-frame-flow as a starting point. We downloaded foldseek clustered AFDB dataset from GENIE2 https://github.com/aqlaboratory/genie2 and processed the data with FrameFlow's process_pdb_files.py script to be in a format usable in the FrameFlow experiment code. To implement DiT, we used https://github.com/facebookresearch/DiT in which we incorporated RoPE with https://github.com/lucidrains/rotary-embedding-torch. Our encoder uses ProteinMPNN's code downloaded from https://github.com/dauparas/ProteinMPNN. Code for this work will be made publicly available on Github with the deanonymized version. We provide sufficient details and references in our work such that our results can be reproduced. Sec. 3 and App. B.1 provide model and training details. Our metrics are defined in App. B.3. Instructions for how each baseline were ran is included in App. C.3.

## 8 ETHICS STATEMENT

We develop a novel method for protein structure generation that can be used in real world protein design applications. Our work is purely academic to advance machine learning techniques for protein data which can be used in down stream applications that are both ethical and unethical. Fortunately most applications with protein design are targeted at developing new drugs and medicines for which the benefits can outweight harmful impact. Protein design is a rapidly developing field with biosecurity becoming a crucial consideration to which we refer to responsiblebiodesign.ai for more detail.

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

# A ADDITIONAL BACKGROUND

Here we provide a formal derivation of eq. (4) based on linear SDEs. Using SDEs for generative modeling can be traced back to Song et al. (2021); Sohl-Dickstein et al. (2015); Ho et al. (2020). Our derivation is not novel and follows the same steps as Song et al. (2021); Zheng et al. (2023). It comprises of two main objects: a forward SDE to corrupt data and a reverse SDE to generate data from noise. The most common SDE for generative modeling is of the Itô form and with linear drift and diffusion coefficients. The forward SDE is defined as

$$\mathrm{d}\mathbf{z}^{(t)} = a(t)\mathbf{z}^{(t)}\mathrm{d}t + b(t)\mathrm{d}\mathbf{w}^{(t)} \tag{7}$$

where $a(t) : [0, 1] \to \mathbb{R}$ and $b(t) : [0, 1] \to \mathbb{R}$ are the drift and diffusion coefficients, respectively, and $\mathbf{w}^{(t)}$ is a Wiener process. The seminal result of Anderson (1982) showed that eq. (7) can be reversed in time analytically with the following reverse SDE

$$\mathrm{d}\mathbf{z}^{(t)} = \left[a(t)\mathbf{z}^{(t)} - b(t)^2\nabla_{\mathbf{z}^{(t)}}\log p_0(\mathbf{z}^{(0)})\right]\mathrm{d}t + b(t)\mathrm{d}\mathbf{w}^{(t)} \tag{8}$$

in the sense that the marginal distributions $p_t(\mathbf{z}^{(t)})$ agree between the two SDEs.

The key idea will be to derive $a(t)$ and $b(t)$ for the forward SDE in eq. (7) that matches the time-dependent noising process in eq. (2). With this, we can plug $a(t)$ and $b(t)$ into the reverse SDE in eq. (8) to generate samples from the latent space. The time derivative of the mean and covariance of eq. (7) at each time $t$ is a result found in Section 5.5 of Särkkä & Solin (2019),

$$\frac{\partial}{\partial t}\log\mathbb{E}[\mathbf{z}^{(t)}] = a(t)$$

$$\frac{\partial}{\partial t}\mathrm{Var}(\mathbf{z}^{(t)}) = 2a(t)\mathrm{Var}(\mathbf{z}^{(t)}) + b(t)^2.$$

From eq. (2), we know the mean and variance: $\mathbb{E}[\mathbf{z}^{(t)}] = \alpha(t)\mathbf{z}^{(0)}$ and $\mathrm{Var}(\mathbf{z}^{(t)}) = \sigma(t)^2 I$. First solving for $a(t)$,

$$a(t) = \frac{\partial}{\partial t}\log\left(\alpha(t)\mathbf{z}^{(0)}\right)$$

$$= \frac{\partial}{\partial t}\log\alpha(t).$$

Next solving for $b(t)$,

$$b(t)^2 = 2a(t)\sigma(t)^2 - \frac{\partial}{\partial t}\sigma(t)^2$$

$$= 2\sigma(t)\left(\frac{\partial}{\partial t}\sigma(t) - a(t)\sigma(t)\right).$$

This matches the form of $a(t)$ and $b(t)$ in eq. (4).

# B  ADDITIONAL METHOD

## B.1  LSD DETAILS

We describe LSD training and neural network architecture details. First, we recall the training objetives and neural networks described in Sec. 3. As a reminder, $L$ is the length of the protein and $K$ is the number of latent dimensions.

1. Encoder $p_\phi$ with weights $\phi$ parameterized as a modified version of ProteinMPNN (Dauparas et al., 2022) to output the mean and variance of the latent distribution instead than amino acid probabilities. The input to the encoder is the protein structure $\mathbf{x} \in \mathbb{R}^{L \times K}$ while the output is the mean $\mu \in \mathbb{R}^{L \times K}$ and log standard deviation $\log \sigma \in \mathbb{R}^{L \times K}$ of the latent distribution $\mathbf{z} \in \mathbb{R}^{L \times K}$. We use a hidden dimension of 128, no dropout, and 6 message passing layers. All other details of ProteinMPNN are kept the same as reported in its original paper.

2. Decoder $p_\psi$ with weights $\psi$ parameterized as a three layer multi-layer perceptron with 128 hidden dimensions and ReLU activations. The input to the decoder is the latent $\mathbf{z}$ while the output is the contact map $\hat{\mathbf{c}}_\psi \in \mathbb{R}^{L \times L}$.

3. Latent Diffusion Model (LDM) $\hat{\mathbf{z}}_\theta$ with weights $\theta$ parameterized as a Diffusion Transformer (DiT) (Peebles & Xie, 2023b). To use DiT for our purposes, we treat each residue as a token. Specifically, since the noisy latent $\mathbf{z}^{(t)}$ is an input to the model, each $\mathbf{z}_i^{(t)}$ for $i \in [1, \ldots, L]$ is a token where $\mathbf{z}^{(t)} = [\mathbf{z}_1^{(t)}, \ldots, \mathbf{z}_L^{(t)}]$. We use 24 DiT blocks with 384 hidden dimension, 0.1 Dropout, and Rotary Positional Encodings (RoPE) (Su et al., 2024) in place of abolsute positional encodings during the attention operations.

4. Structure Diffusion Model (SDM) $\hat{\mathbf{x}}_\varphi$ with weights $\varphi$ parameterized as FrameFlow (Yim et al., 2024a). We use the same hyperparameters as FrameFlow, 256 single dimension and 128 pair dimension, with the addition of concatenating the latents $\mathbf{z}^{(0)}$ and the predicted contact map $\hat{\mathbf{c}}_\psi(\mathbf{z}^{(0)})$ to the initial set of 1D and 2D features provided to FrameFlow. We found rotation annealing, auxiliary losses, and self-conditioning unnecessary and removed them for a simpler model.

Each model is trained with the following losses. We have slighlty modified each loss from its initial presentation in the main text to be more explicit:

**Encoder and decoder loss:**

$$\mathcal{L}_{\text{rec}}(\mathbf{z}, \mathbf{x}) = \frac{1}{|\mathcal{Z}_0|} \sum_{(i,j) \in \mathcal{Z}_0} -\log p_\psi(\mathbf{c}_{ij} = 0 | \mathbf{z}_i \otimes \mathbf{z}_j) + \frac{1}{|\mathcal{Z}_1|} \sum_{(i,j) \in \mathcal{Z}_1} -\log p_\psi(\mathbf{c}_{ij} = 1 | \mathbf{z}_i \otimes \mathbf{z}_j)$$

$$\mathcal{L}_{\text{VAE}}(\mathbf{x}) = \mathbb{E}_{p_\phi(\mathbf{z}|\mathbf{x})} [\mathcal{L}_{\text{rec}}(\mathbf{z}, \mathbf{x})] + \lambda \mathbb{KL} [p_\phi(\mathbf{z}|\mathbf{x}) || \mathcal{N}(0, I)]$$

with $\mathcal{Z}_0 = \{(i,j) : \mathbf{c}_{ij} = 0\}$ and $\mathcal{Z}_1 = \{(i,j) : \mathbf{c}_{ij} = 1\}$ as the set of indices where $\mathbf{c}_{ij} = 0$ and $\mathbf{c}_{ij} = 1$ respectively.

**LDM loss:**

$$\mathcal{L}_{\text{LDM}}(\mathbf{z}) = \mathbb{E}_{\substack{q_t(\mathbf{z}^{(t)}|\mathbf{z}) \\ \mathcal{U}(t;0,1)}} \left[ \frac{1}{\sigma(t)^2} \| \hat{\mathbf{z}}_\theta(\mathbf{z}^{(t)}, t) - \mathbf{z} \|_2^2 \right].$$

**SDM loss:**    Following FrameFlow, we represent the atomic coordinates $\mathbf{x}$ as elements of SE(3) called frames, $\mathbf{T}(\mathbf{x}) \in \text{SE}(3)^L$. For brevity, we will use $\mathbf{T} = \mathbf{T}(\mathbf{x})$. Let $\mathbf{T} = [\mathbf{T}_1, \ldots, \mathbf{T}_L]$ be the $L$ frames of the structure obtained by converting atomic coordinates to frames. Since $\text{SE}(3) = \mathbb{R}^3 \ltimes \text{SO}(3)$, we can represent each frame $\mathbf{T}_i = (\tau_i, \mathbf{R})_i$ for all $i$ by an translation $\tau_i \in \mathbb{R}^3$ and rotation $\mathbf{R}_i \in \text{SO}(3)$. Converting atomic coordinates to the frame representation is achieved by setting the Carbon-alpha coordinate as translation and using the Gram-Schmidt process to construct the orthonormal basis of the remaining residues (Yim et al., 2023). For shorthand, we will use $\mathbf{T} = (\tau, \mathbf{R})$ where $\tau \in \mathbb{R}^{L \times 3}$ and $\mathbf{R} \in \text{SO}(3)^L$. In other words, $\tau$ and $\mathbf{R}$ refers to the translations and rotations of all residues. The SDM's predictions can be written as

$$\hat{\mathbf{x}}_\varphi(\mathbf{x}^{(t)}, \hat{\mathbf{c}}_\psi(\mathbf{z}), \mathbf{z}, t) = \hat{\mathbf{T}}(\mathbf{x}^{(t)}, \hat{\mathbf{c}}_\psi(\mathbf{z}), \mathbf{z}, t) = (\hat{\tau}(\mathbf{x}^{(t)}, \hat{\mathbf{c}}_\psi(\mathbf{z}), \mathbf{z}, t), \hat{\mathbf{R}}(\mathbf{x}^{(t)}, \hat{\mathbf{c}}_\psi(\mathbf{z}), \mathbf{z}, t))$$

We can now write the SDM loss:

$$\mathcal{L}_{\text{trans}}(\mathbf{T}, \mathbf{T}^{(t)}, \mathbf{z}, t) = \frac{\left\| \tau - \hat{\tau}(\mathbf{T}^{(t)}, \hat{\mathbf{c}}_\psi(\mathbf{z}), \mathbf{z}, t) \right\|^2}{\sigma(t)^2}$$

$$\mathcal{L}_{\text{rot}}(\mathbf{T}, \mathbf{T}^{(t)}, \mathbf{z}, t) = \frac{\left\| \log_{\mathbf{R}^{(t)}}(\mathbf{R}) - \log_{\mathbf{R}^{(t)}}(\hat{\mathbf{R}}(\mathbf{T}^{(t)}, \hat{\mathbf{c}}_\psi(\mathbf{z}), \mathbf{z}, t)) \right\|^2}{\sigma(t)^2}.$$

$$\mathcal{L}_{\text{SDM}}(\mathbf{T}, \mathbf{z}) = \mathbb{E}_{\substack{q_t^*(\mathbf{T}^{(t)}|\mathbf{T}) \\ \mathcal{U}(t;0,1)}} \left[ \mathcal{L}_{\text{trans}}(\mathbf{x}, \mathbf{x}^{(t)}, \mathbf{z}, t) + \mathcal{L}_{\text{rot}}(\mathbf{x}, \mathbf{x}^{(t)}, \mathbf{z}, t) \right]$$

where $q_t^*(\mathbf{T}^{(t)}, \mathbf{T}) = [\Phi_t]_* q_0(\mathbf{T}^{(1)})$ is defined with the prior $q_0 = \mathcal{U}(\text{SO}(3))^L \times \mathcal{N}(0, 1)^{L \times 3}$ and push-forward using the conditional flow $\Phi_t(\mathbf{T}^{(1)}|\mathbf{T}^{(0)}) = \mathbf{T}^{(t)} = [\mathbf{T}_1^{(t)}, \ldots, \mathbf{T}_L^{(t)}]$ where $\mathbf{T}_i^{(t)} = (\tau_i^{(t)}, \mathbf{R}_i^{(t)})$ defined as the geodesics

$$\tau_i^{(t)} = (1-t)\tau_i^{(0)} + t\tau_i^{(1)}, \qquad \mathbf{R}_i^{(t)} = \exp_{\mathbf{R}_i^{(1)}} \left( (1-t)\log_{\mathbf{R}_i^{(1)}}(\mathbf{R}_i^{(0)}) \right).$$

$\exp$ and $\log$ refer to the exponential and logarithm map onto the respective manifolds. For more details of the SDM training, we refer to (Yim et al., 2024a).

**Multi-stage training.** We use three stages of training as described in Sec. 3.3. In stage 1, the VAE is trained. In stage 2, the SDM is trained and the VAE is fine-tuned jointly with the SDM. In stage 3, the LDM is trained with the VAE weights fixed. A summary of the training stages, losses and number of epochs is provided in Tab. 3. Each stage uses the AdamW optimizer (Loshchilov, 2017) with learning rate 1e-4 and weight decay 1e-5. We trained on 8 Nvidia A6000 GPUs for each stage. We used the length-based mini-batching strategy from (Yim et al., 2023) that came with the FrameFlow codebase.

Table 3: Training stages.

| Stage | Loss | Epochs/Days |
|---|---|---|
| 1: VAE training | $\mathbb{E}_{p(\mathbf{x})}[\mathcal{L}_{\text{VAE}}(\mathbf{x})]$ | 16/0.5 |
| 2: VAE & SDM training | $\mathbb{E}_{p(\mathbf{x})}\left[\mathcal{L}_{\text{VAE}}(\mathbf{x}) + \mathbb{E}_{p_\phi(\mathbf{z}|\mathbf{x})}[\mathcal{L}_{\text{SDM}}(\mathbf{T}(\mathbf{x}), \mathbf{z})]\right]$ | 16/1 |
| 3: LDM training | $\mathbb{E}_{p(\mathbf{x}),p_\phi(\mathbf{z}|\mathbf{x})}[\mathcal{L}_{\text{LDM}}(\mathbf{z})]$ | 48/2 |

## B.2 PAE GUIDANCE DETAILS

The training dataset is constructed by sampling 500 proteins of each length in the range 60 to 128 from AFDB. ProteinMPNN samples three sequences per backbone and AlphaFold2 in single sequence mode is used to compute a mean PAE value per sequence. The minimum mean PAE value amongst the sequences for each backbone is used as the corresponding label. A min-max norm is used to transform the PAE values to lie between 0 and 1. To parameterize the regressor we use two 1D convolutional layers with kernel size $k = 5$ and 256 channels. Following each convolutional layer, ReLU activation and dropout $p = 0.2$ are applied. An attention pooling mechanism aggregates the embeddings across the length dimension, and a linear projection transforms the fixed length embedding to a single dimension. While Dhariwal & Nichol (2021) propose training the guide function on noisy $\mathbf{z}^{(t)}$ samples, we found $p_t(\mathbf{y}|\mathbf{z}^{(t)})$ difficult to learn as seen in the Table 4 below.

Table 4: Guide model ablation.

| Input | PearsonR |
|---|---|
| Noised latents | 0.34 |
| Denoised latents | 0.40 |

Since the LDM is tasked with predicting denoised latents, $\mathbf{z}^{(0)}$, we can train with the following L2 loss:

$$\mathcal{L} = \|\hat{\mathbf{z}}_\theta(\mathbf{z}^{(t)}, t) - \mathbf{y}\|_2^2$$

where $\mathbf{y} \in \mathbb{R}$ is the designed PAE label. Models were trained on 2 A100s for 12 hours, and the best checkpoint was selected by computing PearsonR on a held-out set of designed backbones from the PDB. Sweeps over $\omega_{\text{PAE}}$ from 0-200 across proteins of length 75, 100, and 125 demonstrate the ability of PAE guidance to reduce mean PAE of generated samples evaluated with Alphafold2 see Figure 6.

### B.3 METRIC DETAILS

We describe each metric used in Sec. 5 for completeness. Designability, diversity, and novelty are standard metrics used in multiple prior related works (Yim et al., 2023; Watson et al., 2022; Lin et al., 2024; Bose et al., 2024; Huguet et al., 2024). Designable Pairwise TM-score and secondary structure composition are an additional metric reported in recent works as well (Lin et al., 2024; Bose et al., 2024; Huguet et al., 2024). We introduce Secondary Structure Distance to supplement the above metrics. Below we describe each metric.

1. **Designability** (Des): Let $\mathbf{x}$ be a protein backbone structure sampled from a protein structure generative model. We use the open-sourced ProteinMPNN code[4] to generate 8 sequences for each backbone. ESMFold (Lin et al., 2023) then predicts the structure of each sequence. We compute the atomic Root Mean Squared Deviation (RMSD) of each *predicted* structure against the *sampled* structure $\mathbf{x}$. If RMSD $< 2.0$ then we consider $\mathbf{x}$ to be *designable* in the sense that a sequence can be found which would fold into the $\mathbf{x}$ structure. Clearly this evaluation is purely in-silico and only serves as a approximation of whether a structure is designable. Despite that, designability has been found to correlate with wet-lab success especially when more specialized sequence and structure prediction models are used (Watson et al., 2022; Zambaldi et al., 2024). We report designability as the percentage of samples that are designable. It is currently debated whether a protein generative model should aim to have as high designability as possible since designability is influenced by the inductive biases of the structure prediction model. Many natural occurring proteins are known to not pass the designability crition (Huguet et al., 2024; Campbell et al., 2024) yet are real proteins. We present our experiments with designability as a metric we wish to increase in order to follow prior works.

2. **Diversity** (Div): A generative model can achieve $100\%$ designability by repeatedly sampling the same designable structure. To detect this exploitation, we report diversity as the number of clusters after running a clustering algorithm over the designable samples. Following prior works starting with (Trippe et al., 2022), we MaxCluster (Herbert & Sternberg, 2008) to run hierarchical clustering over all designable structures with average linkage, 0.5 TM-score cutoff, and no sequence filtering. The goal is to maximize the number of clusters in the samples.

3. **Designable Pairwise TM-score** (DPT): Reporting the number of clusters can be biased since there are many hyperparameters and algorithms for clustering. To present a unbiased view of diverisy, we report the average TM-score of all the pairwise TM-scores between designable samples. This is part of an auxiliary output after running MaxCluster.

4. **Novelty**: We measure how a method extrapolates beyond the training set by computing the average of the maximum TM-scores of each designable structure $\mathbf{x}$ when compared to AFDB. We use FoldSeek with the following command: `foldseek easy-search <path-to-designable-pdb-files> <path-to-afdb-database> alignments.m8 tmp --alignment-type 1 --format-output query,target,alntmscore,lddt --tmscore-threshold 0.0 --exhaustive-search --max-seqs 10000000000 --comp-bias-corr 0 --mask 0`. Foldseek commands are chosen to ignore sequence filtering and turn off pre-filtering steps before running a efficent structural search algorithm against all structures in the AFDB. The goal is to minimize the novelty metric as this corresponds to more extrapolation beyond the training set while adhering to the designability criterion. We note

---

[4]https://github.com/dauparas/ProteinMPNN

this is just *one* definition of novelty and other definitions as possible but we aim to follow precedent set by prior works.

5. **(Designable) Helix/Strand/Coil percentage** (H/S/C): Out of the designable samples, we use DSSP (Kabsch & Sander, 1983) to compute the average alpha-helix, beta-strand, and coil percent the set of designable samples. Previous works have noted protein generative models are susceptible to preferring helices over strands (Huguet et al., 2024). This metric helps see how diverse the designable structures are from a secondary structure perspective. There is no inherently desirable value of this metric but serves to provide intuition into how biased a model is to sampling helices over strands.

6. **Secondary Structure Distance** (SSD): As mentioned in Sec. 5.1, SSD is meant to supplement the above metrics since none of them address how well a protein generative model is learning the training distribution. For instance, datasets such as PDB and AFDB have less than 100% designability; a generative model trained on these datasets cannot achieve 100% designability if they learn the datasets perfectly. There is a need for a distributional metric that measures how well a generative model captures the training distribution. We propose SSD to provide insight into how well a generative model learns the secondary structure distribution of the training dataset. The goal is to lower SSD as that corresponds to a lower distributional distance between the alpha and helical composition of the training set and the generated samples. Unlike the above metrics, we compute SSD over all the generated samples without filtering for designability for the reasons just discussed. Computation of SSD is provided in Alg. 1.

---

**Algorithm 1** Computation of the Secondary Structure Distance (SSD) Metric

---

1: **Input:** Training dataset $\mathcal{D}_{\text{train}}$, Generated samples $\mathcal{D}_{\text{gen}}$
2: $\mathcal{D}_{\text{train\_sampled}} \leftarrow$ Random Sample 10,000 proteins from $\mathcal{D}_{\text{train}}$
3: **for** $\mathcal{D} \in \{\mathcal{D}_{\text{train\_sampled}}, \mathcal{D}_{\text{gen}}\}$ **do**
4:     Initialize set $\text{SS}_{\mathcal{D}} \leftarrow \emptyset$
5:     **for** protein $p$ in $\mathcal{D}$ **do**
6:         Compute helix percentage $P_H^{(p)}$ and strand percentage $P_S^{(p)}$
7:         Store $(P_S^{(p)}, P_H^{(p)})$ in $\text{SS}_{\mathcal{D}}$
8:     **end for**
9: **end for**
10: Divide $[0, 1]$ into $n$ equal bins $B_1, B_2, \ldots, B_n$
11: **for** $\mathcal{D} \in \{\text{SS}_{\text{train\_sampled}}, \text{SS}_{\text{gen}}\}$ **do**
12:     **for** $i = 1$ to $n$ **do**
13:         **for** $j = 1$ to $n$ **do**
14:             $P_{\mathcal{D}}(i, j) \leftarrow \frac{|\{(P_S, P_H) \in \text{SS}_{\mathcal{D}} \mid P_S \in B_i \wedge P_H \in B_j\}|}{|\text{SS}_{\mathcal{D}}|}$
15:         **end for**
16:     **end for**
17: **end for**
18: $W \leftarrow \text{WassersteinDistance}(P_{\text{train\_sampled}}, P_{\text{gen}})$
19: **return** $W$

---

## C    ADDITIONAL EXPERIMENTS

### C.1    LSD EXPERIMENTS

To find the optimal hyperparameters for the VAE, we performed a hyperparameter sweep over the number of latent dimensions $K$ and the regularization weight $\lambda$. For each combination of $K \in \{2, 4, 8\}$ and the regularization weight $\lambda \in \{0.01, 0.1, 1.0\}$, we ran stage 1 training followed by evaluating long range contact ROC and PRAUC on a held out set of randomly chosen 32 protein clusters using Foldseek's cluster assignment. The results are presented in Tab. 5 where we see $K = 4$ and $\lambda = 0.01$ to be optimal.

Table 5: VAE hyperparameter sweep.

| $K$ | $\lambda$ | ROCAUC | PRAUC |
|---|---|---|---|
| 2 | 0.01 | 0.5 | 0.5 |
| 2 | 0.1 | 0.5 | 0.5 |
| 2 | 1.0 | 0.5 | 0.51 |
| **4** | **0.01** | **0.99** | **0.99** |
| 4 | 0.1 | 0.5 | 0.5 |
| 4 | 1.0 | 0.5 | 0.5 |
| 8 | 0.01 | 0.99 | 0.99 |
| 8 | 0.1 | 0.99 | 0.99 |
| 8 | 1.0 | 0.5 | 0.5 |

Using $K = 4$ and $\lambda = 0.01$, we ran the full three stage training procedure for different ablations of the LDM in Tab. 6. We report the standard Designability, Diversity, and Novelty metrics where we find using DiT and RoPE presented the best combined of the metrics.

Table 6: Ablations

| Ablation | Des | Div | Nov |
|---|---|---|---|
| DiT+RoPE | **68.7** % | 203 | 0.74 |
| DiT No RoPE | 57.25% | **212** | 0.74 |
| Transformer instead of DiT | 51.88% | 190 | **0.73** |

Next, we investigated if (1) separate training of the autoencoder and FrameFlow is necessary and (2) if the contact map loss is necessary. We trained three different settings:

1. End-to-end training of autoencoder and FrameFlow from scratch for 32 epochs **without** contact map loss.

2. End-to-end training of autoencoder and FrameFlow from scratch for 32 epochs **with** contact map loss.

3. Two stage training of autoencoder and FrameFlow as defined in Tab. 3 with 16 epochs for each stage.

Tab. 7 shows the results after training where we evaluated the average reconstruction RMSD on the autoencoder validation set. Specifically, we encoded each protein in the validation set then sampled with FrameFlow to reconstruction the protein. We take the RMSD of the sampled protein against the encoded protein and report the average RMSD across all examples. We clearly see that the two stage training with contact map loss results in the lowest RMSD. Hence, both the contact map loss and two stage training are necessary.

With $K = 4$ and $\lambda = 0.01$ and our DiT+ROPE LDM, we sampled at different values of $\gamma$ and tried the noiseless ODE formulation of sampling. We find $\gamma = 0.7$ provided the sweet spot of highest

diversity with good designability. As discussed in App. B.3, one should not always try to maximize designability but also consider diversity and novelty.

We show scRMSD results across different lengths for each variant of LSD in Fig. 8. We perform analysis of dihedral angles in Fig. 9 between samples from LSD and the training dataset.

Table 7: Autoencoder training ablations.

| Training | Contact map loss | Reconstruction RMSD |
|---|---|---|
| End-to-end training of autoencoder and FrameFlow from scratch. | No | 12.0 |
| | Yes | 4.1 |
| Two stage training of autoencoder then FrameFlow as done in Tab. 3. | Yes | 1.8 |

Table 8: $\gamma$ hyperparameter sweep.

| $\gamma$ | | **Des** ($\uparrow$) | **Div** ($\uparrow$) | **Nov** ($\downarrow$) | **SCC** ($\downarrow$) |
|---|---|---|---|---|---|
| 0.5 | SDE | **89.4**% | 148 | 0.78 | 1.094 |
| 0.6 | SDE | 81.1% | 197 | 0.75 | 1.036 |
| 0.7 | SDE | 68.7% | **203** | 0.74 | 0.859 |
| 0.8 | SDE | 46.2% | 164 | 0.72 | 0.589 |
| 0.9 | SDE | 30.9% | 130 | 0.72 | 0.398 |
| 1.0 | SDE | 15.3% | 84 | **0.70** | **0.132** |
| | ODE | 5.5% | 20 | 0.75 | 0.608 |

Table 9: AFDB Metrics. We took 10 random proteins across each length between 60-128 and evaluated designability and diversity. This demonstrates the low designability but high diversity nature of AFDB.

| | **Des** ($\uparrow$) | **Div** |
|---|---|---|
| AFDB | 12.9% | 453 |

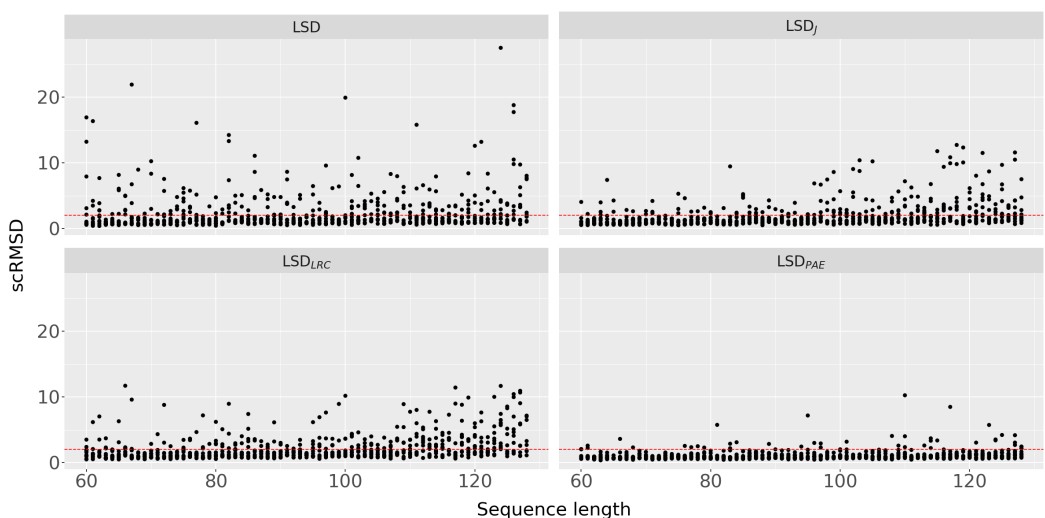

Figure 8: scRMSD plotted against sample length for all variants of our LSD with $\gamma = 0.7$.

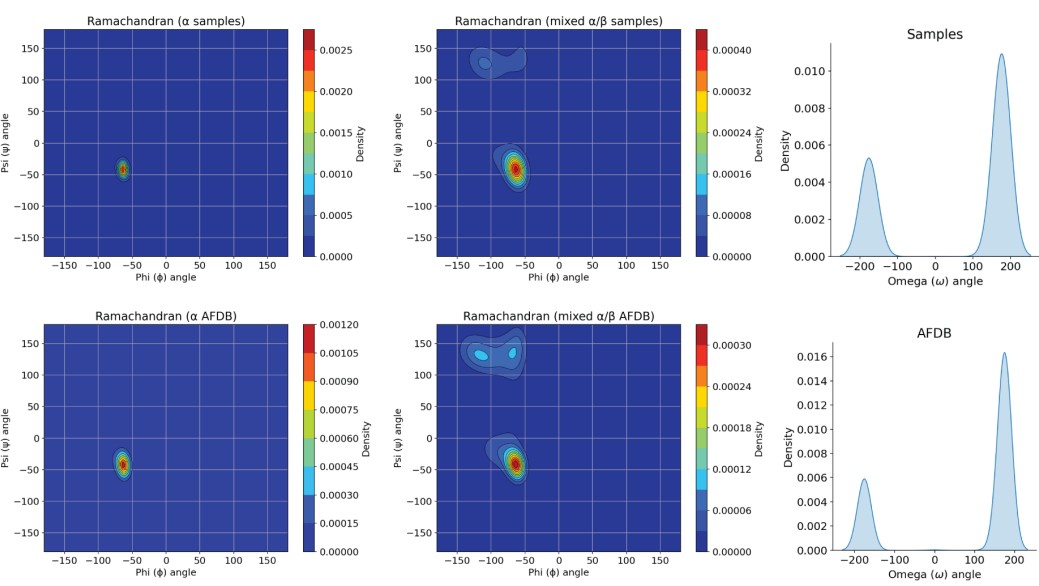

Figure 9: Ramachandran plot of LSD ($\gamma = 0.7$) samples from Sec. 5.4 compared to 1000 structures randomly sampled from the AFDB training set. For visualization purposes, we separated Ramachandran plots between $\alpha$-helical and mixed $\alpha$-helical/$\beta$-sheet samples. In the last column we plot the $\omega$ dihedral angle. We find LSD and AFDB have very similar dihedral angle distributions.

### C.2 LSD GUIDANCE EXPERIMENTS

Similar to Tab. 1, we sweep hyperparameters $\omega_{PAE}$, $r_{LRC}$ at $\gamma = 0.7$ to find the best setting for each model. Our results are shown in Tab. 10. We selected hyperparameters with the following logic:

- LSD$_{PAE}$ ($\omega_{PAE}$=25): We increased $\omega_{PAE}$ until designability was maximized without decreasing diversity.
- LSD$_{LRC}$ ($r_{LRC}$=1): We selected the weight with the best novelty.
- LSD$_J$ ($\omega_{PAE}$=5, $r_{LRC}$=5): We selected the weights that maximized diversity.

that maximized designability without sacrificing diversity for LSD$_{PAE}$ ($\omega_{PAE}$=25);

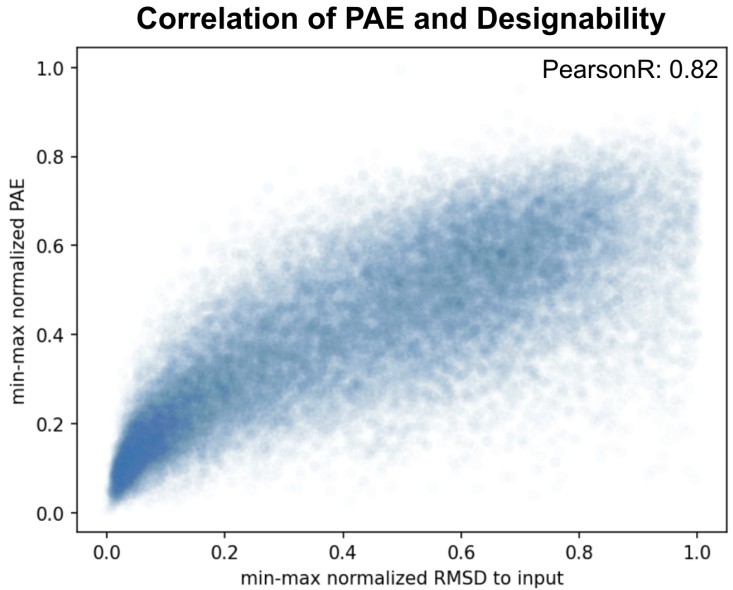

Figure 10: Correlation of normalized rmsd to input and mean PAE values.

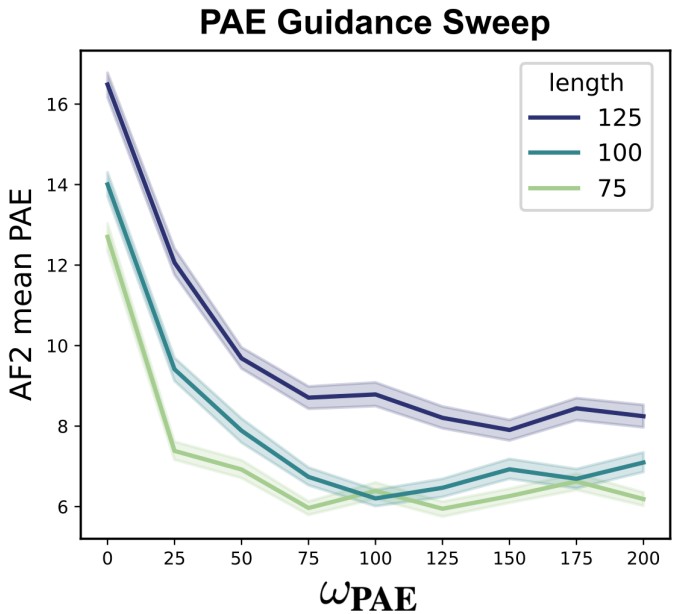

Figure 11: Samples at $\gamma=1$ with different guidance scales. Increasing $\omega_{\mathrm{PAE}}$ leads to lower mean PAE of sampled structures from Alphafold2.

### C.3 BASELINE EXPERIMENTS

To facilitate a comprehensive comparison with our proposed method, we evaluated several protein generative models. This section details the procedures followed, including the selection of hyperparameters.

Table 10: LSD results with $\gamma = 0.7$. We use 100 timesteps for both latent and structure diffusion.

| Method | $\omega_{\mathbf{PAE}}$ | $r_{\mathbf{LRC}}$ | Des ($\uparrow$) | Div ($\uparrow$) | DPT ($\downarrow$) | Nov ($\downarrow$) | SSD ($\downarrow$) | H/S/C |
|---|---|---|---|---|---|---|---|---|
| LSD | | | 69% | 203 | 0.46 | 0.74 | 0.86 | 76/4/20 |
| LSD$_{\mathrm{PAE}}$ | 50 | | 95% | 176 | 0.4 | 0.7 | 1.06 | 76/5/20 |
| | 25 | | **94**% | 204 | **0.42** | 0.71 | 1.03 | 75/4/20 |
| | 10 | | 88% | 211 | 0.43 | 0.73 | 0.95 | 76/4/20 |
| | 5 | | 78% | 193 | 0.43 | 0.74 | 0.87 | 75/4/21 |
| LSD$_{\mathrm{LRC}}$ | | 10 | 73% | 240 | 0.49 | 0.69 | 0.47 | 64/11/25 |
| | | 5 | 67% | 272 | 0.53 | 0.65 | 0.22 | 55/16/29 |
| | | 1 | 33% | 182 | 0.59 | **0.61** | 0.24 | 44/21/35 |
| LSD$_{\mathrm{J}}$ | 10 | 5 | 76% | 262 | 0.49 | 0.65 | 0.26 | 60/13/27 |
| | 5 | 5 | 74% | **296** | 0.52 | 0.65 | 0.26 | 57/15/28 |
| | 10 | 1 | 48% | 232 | 0.56 | **0.61** | 0.47 | 47/20/33 |
| | 5 | 1 | 38% | 265 | 0.54 | **0.61** | **0.23** | 46/20/23 |

### C.3.1 ESM3 UNCONDITIONAL GENERATION

Following Appendix A.3.6 of ESM3, which outlines the procedure for unconditional generation, we employed the open-source 1.4B parameter model available at https://github.com/evolutionaryscale/esm.

To generate a protein sequence of length $l$, we input a sequence of mask tokens of the same length into the model. We set the temperature to 0.5 and configured the number of decoding steps to equal the protein length $l$. Subsequently, we conditioned the 1.4B model to generate structure tokens using the same number of decoding steps ($l$) with argmax decoding, where the temperature was set to 0.

We conducted ablation studies to determine the optimal number of decoding steps by testing values of $l/2$, $2l/3$, and $l$ with a temperature of 0.7. Our experiments revealed that setting the number of decoding steps to $l$ yielded higher diversity than $2l/3$ with slightly lower designability . Additionally, we performed ablation on the temperature parameter by evaluating temperatures of 0.3, 0.5, and 0.7 for the optimal number of decoding steps $l$. We found that a temperature of 0.5 provided the best results.

Table 11: Ablation on Decoding Steps with Fixed Temperature (0.7)

| Decoding Steps | Des ($\uparrow$) | Div ($\uparrow$) | Novelty ($\downarrow$) |
|---|---|---|---|
| l/2 | 23.1% | 62 | 0.83 |
| 2l/3 | 28.5% | 72 | 0.84 |
| l | 26 % | 78 | 0.82 |

Table 12: Ablation on Temperature with Fixed Decoding Steps ($l$)

| Temperature | Des ($\uparrow$) | Div ($\uparrow$) | Novelty ($\downarrow$) |
|---|---|---|---|
| 0.3 | 44.8% | 37 | 0.9 |
| 0.5 | 40.3% | 84 | 0.87 |
| 0.7 | 26 % | 72 | 0.82 |
| 1.0 | 14.5 % | 41 | 0.81 |

To verify our baseline results for the ESM3 model, we consulted with the ESM3 authors. They recommended employing a chain-of-thought experiment that involves sampling the secondary structure with a temperature of 0.7, followed by sampling structure tokens with the same temperature. This approach significantly enhances designability. Consequently, the ESM3 results presented in the main text are based on this approach.

### C.3.2   GENIE 2

To generate samples using Genie2, we utilize the open-source repository available at `https://github.com/aqlaboratory/genie2`. We used a sampling temperature of 0.6. This hyperparameter was selected based on the paper's findings, which indicated that these settings yielded the best performance in terms of sample designability and diversity.

### C.3.3   LATENTDIFF

We benchmarked Latentdiff model using the open-source code available at `https://github.com/divelab/AIRS/tree/main/OpenProt/LatentDiff`. However, due to stochastic sampling of the length in the reconstruction process, we were unable to control the lengths of the sampled proteins. During the upsampling stage in the decoder this MLP was used to process the final node embeddings and predict whether a reconstructed node corresponds to a padded node, this introduced stochasticity in the lengths of the sampled proteins restricting our ability to generate specific lengths reliably especially those longer than 100.

