# OpenReview forum: "Hierarchical Protein Backbone Generation with Latent and Structure Diffusion"
_ICLR.cc/2025/Conference — Submitted to ICLR 2025_

### Official Review · Reviewer_S3eE · 2024-10-29

**Soundness:** 3
**Presentation:** 3
**Contribution:** 2
**Rating:** 3
**Confidence:** 4

**Summary:**

**There is a pentail violation of the anonymity policy in this paper**

This paper leverages the diffusion model to generate protein structures. Sampling is conducted on the level of atomic coordinates. The high-level guidance from the protein backbone presents the effects on generation. The experiment is conducted on AFDB to verify its effectiveness.

**Strengths:**

1. The paper has good quality and the proposed method looks working well.
2. It is novel to sample on the level of acid coordinates.
3. The experiment is conducted on AFDB and various performance metrics are reported.

**Weaknesses:**

1. **Violation of the anonymity policy.** It looks like the author changed the style file in the template. The references and table numbers are highlighted, which makes the paper stand out from other submissions. It is not fair for other submissions.

2. The paper proposes to sample the atomic coordinates. However, it is not clear how to guarantee that the sampled coordinates can form a possible structure on a higher level in practice. The constraint between the two levels is missing in the training process. Other important property such as SE3 is not discussed and addressed in the method.

3. Important theoretical proof is missing. It looks like the paper intuitively leverages the reverse process of the diffusion model on the sequence. The original diffusion models such as DDPM or DDIM conduct the forward and the reverse process on the same level, which makes solving the Markov process tracable. However, it is not proven it still works when applied to two different representations.

4. Quality of experiment. The paper only introduces **only one dataset** in the experiment. It is not conclusive. Please conduct the experiment on all benchmarks in [1].
[1] ProteinGym: Large-Scale Benchmarks for Protein
Fitness Prediction and Design

5. Hierarchical Protein generation is already proposed by existing works, which may limit the contribution of this work. The author may want to integrate more baselines such as [1]
[1]	Zhilin Huang, Ling Yang, Xiangxin Zhou, Chujun Qin, Yijie Yu, Xiawu Zheng, Zikun Zhou, Wentao Zhang, Yu Wang, Wenming Yang:
Interaction-based Retrieval-augmented Diffusion Models for Protein-specific 3D Molecule Generation. ICML 2024
[2] Multi-level Protein Structure Pre-training via Prompt Learning
[3] 	Yan Wang, Lihao Wang, Yuning Shen, Yiqun Wang, Huizhuo Yuan, Yue Wu, Quanquan Gu:
Protein Conformation Generation via Force-Guided SE(3) Diffusion Models. ICML 2024
[4] Alex Morehead, Jeffrey Ruffolo, Aadyot Bhatnagar, Ali Madani:
Towards Joint Sequence-Structure Generation of Nucleic Acid and Protein Complexes with SE(3)-Discrete Diffusion.

**Questions:**

Please refer to the weakness.

---

> ### Author Response · Authors · 2024-11-18
> **Response part 1**
>
> We thank reviewer S3eE for their time and review of our work. We appreciate the acknowledgement of the method’s quality and performance. We believe we have addressed all weaknesses and questions. If the reviewer agrees we would very much appreciate a score increase. If not, we are happy to continue discussing any remaining weaknesses and questions. Below we address the reviewer’s comments and concerns.
>
> > Violation of the anonymity policy…
>
> We are unsure what violation the reviewer is referring to. We did not change the style file. It is common practice to highlight table numbers. There are no instructions or commands in the style guide regarding reference highlighting. Furthermore, we are unsure of how either of these would violate anonymity or result in unfair practices.
>
> > The paper proposes to sample the atomic coordinates. However, it is not clear how to guarantee that the sampled coordinates can form a possible structure on a higher level in practice. The constraint between the two levels is missing in the training process.
>
> We are unsure what this question is asking and what “form a possible structure on a higher level in practice” means. We use FrameFlow to sample protein structures conditioned on latents from another latent diffusion model. FrameFlow is already known to sample real proteins. We show empirically by evaluating against prior works that our approach samples realistic, diverse, and novel protein structures.
>
> > Other important property such as SE3 is not discussed and addressed in the method.
>
> We have added line 188 that addresses the equivariant properties. SE3 equivariance is addressed by using FrameFlow which was previously shown to already be SE3 equivariant. This is not a contribution of our work and thus we refer to [1] for more details. Latent diffusion does not require addressing equivariance since we encode structures with ProteinMPNN which is SE3 invariant.
>
> > Important theoretical proof is missing. It looks like the paper intuitively leverages the reverse process of the diffusion model on the sequence. The original diffusion models such as DDPM or DDIM conduct the forward and the reverse process on the same level, which makes solving the Markov process tracable. However, it is not proven it still works when applied to two different representations.
>
> That is incorrect. Please see figure 1. We propose to use a latent diffusion model and a structure diffusion model to sample protein backbone structures. Latent diffusion follows [2] and structure diffusion follows FrameFlow [1] which are both theoretically justified. We do not perform diffusion on the sequence and diffusion is always performed on the save “level” (we are unsure about this terminology).
>
> > Quality of experiment. The paper only introduces only one dataset in the experiment. It is not conclusive. Please conduct the experiment on all benchmarks in [1]. [1] ProteinGym: Large-Scale Benchmarks for Protein Fitness Prediction and Design
>
> Our experiments follow previous works on protein backbone generation [1,3,4]. We use the same AFDB dataset as [3,4]. Using other datasets would make it unfair to compare our method to other baselines. The ProteinGym dataset is for sequence design and has no relevance to structure. Therefore, we are unsure why the reviewer is asking us to compare with ProteinGym.
>
> > Hierarchical Protein generation is already proposed by existing works, which may limit the contribution of this work. The author may want to integrate more baselines such as...
>
> Out of the listed works, they are either not used for protein generation or are not hierarchical in the way the reviewer claims.
> * **Interaction-based Retrieval-augmented Diffusion Models for Protein-specific 3D Molecule Generation**. This work is for small molecule generation. Our work is for protein generation.
> * **Multi-level Protein Structure Pre-training via Prompt Learning**. This work is on pre-training and representation learning. Not for protein generation.
> * **Protein Conformation Generation via Force-Guided SE(3) Diffusion Models**. This work focuses on a different task of conformation generation. They do not apply latent diffusion and do not do hierarchical generation.
> * **Towards Joint Sequence-Structure Generation of Nucleic Acid and Protein Complexes with SE(3)-Discrete Diffusion**. This work does joint sequence and structure generation which is not hierarchical. Furthermore, they are focused on nucleic acid or complex generation. A different task than ours which would be unfair to compare against.
>
> We have to the best of our knowledge listed the known hierarchical and/or latent diffusion models in our related works. Furthermore, many of the papers listed are from ICML 2024 which according to the [ICLR guidelines](https://iclr.cc/Conferences/2025/FAQ) are concurrent works. We ask the reviewer to only suggest possible related works that follow the guidelines.

---

> > ### Author Response · Authors · 2024-11-18
> > **Response part 2**
> >
> > # References
> >
> > [1] Yim, Jason, et al. "Improved motif-scaffolding with se (3) flow matching." _arXiv preprint arXiv:2401.04082_ (2024).
> >
> > [2]  Rombach, Robin, et al. "High-resolution image synthesis with latent diffusion models." _Proceedings of the IEEE/CVF conference on computer vision and pattern recognition_. 2022.
> >
> > [3] Bose, Avishek Joey, et al. "Se (3)-stochastic flow matching for protein backbone generation." _arXiv preprint arXiv:2310.02391_(2023).
> >
> > [4] Lin, Yeqing, and Mohammed AlQuraishi. "Generating novel, designable, and diverse protein structures by equivariantly diffusing oriented residue clouds." arXiv preprint arXiv:2301.12485 (2023).

---

> ### Comment · Reviewer_S3eE · 2024-11-18
> **Response to rebuttal**
>
> Thanks for the rebuttal. Some of my concerns have not been addressed and new concerns have been observed.
>
> W2: When you use FrameFlow to sample real protein structures. how do you guarantee the newly generated protein is still a real protein? If it is a real protein, what is the point of protein generation (i.e., why do we need to generate an existing real protein)?
>
> W3: [1] discuss SE3 with a general diffusion model. While the author claims the proposed work is a hierarchical framework, SE3 on multiple levels should be discussed.
>
> W4: I understand [3,4] use one dataset. But it does not mean [3,4] are high-quality papers. The experiment is not conclusive with one dataset. This can not changed no matter how many papers using AFDB.
>
> W5: There is no paper that can directly be used as baselines without any changes. All these references can be easily introduced as the baselines. Note that the target of adding more baselines is used to improve the experiment quality, not SOTA a specific task.

---

> > ### Author Response · Authors · 2024-11-23
> >
> > > W2: When you use FrameFlow to sample real protein structures. how do you guarantee the newly generated protein is still a real protein?
> >
> > We use the "designability" in-silico metric as a proxy to test whether a generated structure is real. This is described in Appendix B.3 of our work. To summarize, each generated structure is put into ProteinMPNN to sample eight sequences followed by predicting the structure of each sequence with ESMFold to test if any of the ESMFold structure have RMSD < 2 with the generated structure. This metrics was the primary in-silico metric used in the Nature publication RFdiffusion [1]. This round trip RMSD is known to be a successful filter for certain binder design tasks [1, 2, 3, 4] -- note [1, 2, 3] are published in journals and [4] is AlphaProteo from Deepmind. This metric has also been used in previous in-silico papers that have been published at machine learning conferences [5, 6, 7].
> >
> > Our work is purely in-silico. We emphasize that protein design papers at machine learning conference almost never have wet lab validation [5, 6, 7]. The goal is to develop and share novel machine learning techniques that may be translated into wet-lab applications. An example of this is FrameDiff [5] which introduced a machine learning method with purely in-silico results that was used in RFdiffusion (with the FrameDiff first authors) where the method used with wet-lab validation [1].
> >
> > > If it is a real protein, what is the point of protein generation (i.e., why do we need to generate an existing real protein)?
> >
> > Protein generation is a fundamental step in protein design, a field which has been recognized in this year's Nobel prize for chemistry due to the advancements made by AI [8]. De novo protein design aims to generate novel proteins for new functions that we can't find in nature. We include a novelty metric in our work to measure how similar our generated proteins are to known protein structures. We only take proteins that pass the "designability" test (see above) when computing novelty to account for realism. Novelty is reported as a percentage where 1.0 ( i.e. 100%) indicates all our proteins have an exact match in the AFDB based on  the TM-score metric. Our approach $LSD_{LRC}$ achieves a novelty of 0.61 indicating that our approach discovers novel proteins that are dissimilar to known proteins but not completely.
> >
> > The need for novelty is subjective and differs between applications. A unique selling point of our method compared to previous approaches is that we can adjust novelty with our guidance techniques. For instance, using LRC guidance gives 0.61 novelty while not using any guidance gives 0.74, in other words generated proteins are more similar to known proteins. We are happy to answer more questions or go in depth with novelty. A description of novelty is provided in Appendix B.3.
> >
> > Lastly, [1, 2, 3, 4] have shown novel AI generated proteins can be experimentally validated binders with higher success rate than ever before. Therefore, there are real world implications to aid in drug or therapeutic design with AI-generated protein binders.
> >
> > > W3: [1] discuss SE3 with a general diffusion model. While the author claims the proposed work is a hierarchical framework, SE3 on multiple levels should be discussed.
> >
> > To clarify, the referenced work Yim et al. [9] does *not* discuss SE3 with a *general* diffusion model. It discusses SE3 flow matching and SE3 equivariance with FrameFlow which is the method we use as the structure diffusion step. The first step of our method, latent diffusion, is on a latent space that is SE3 invariant also due to prior work.
> > The encoder we use in latent diffusion is ProteinMPNN which uses the same network architecture as Ingraham et al [10] where in section 2.1 it states "the features are invariant to rotations and translations". This is equivalent to being SE3 invariant.
> >
> > Once we sample latents, they are passed onto to the structure diffusion model as conditioning variables. The latents do not possess any SE3 symmetry and hence do not affect the equivariance properties of the structure diffusion. As a result, SE3 symmetries only applies to the structure diffusion step which is already known to be SE3 equivariant due to [9].

---

> > > ### Author Response · Authors · 2024-11-23
> > >
> > > > W5: There is no paper that can directly be used as baselines without any changes. All these references can be easily introduced as the baselines. Note that the target of adding more baselines is used to improve the experiment quality, not SOTA a specific task.
> > >
> > > We respectfully disagree that "all these references can be easily introduced as the baselines". Each of the references focus on a different task that would constitute a new paper. Had our goal been to develop a method for multiple tasks, then we would have done our work differently. We choose to focus on one fundamental task in protein structure modeling and develop a method that improves upon the weaknesses of previous approaches. Protein structure generation is not solved since we are unable to accurately model AFDB and control the properties of generated proteins. Therefore, it is of value to focus on this rather than attempt to be too broad and lose focus.
> > >
> > > For instance, the referenced works
> > > * "Protein Conformation Generation via Force-Guided SE(3) Diffusion Models"
> > > * "Interaction-based Retrieval-augmented Diffusion Models for Protein-specific 3D Molecule Generation"
> > > * "Towards Joint Sequence-Structure Generation of Nucleic Acid and Protein Complexes with SE(3)-Discrete Diffusion"
> > >
> > > each focus on a single task, conformation sampling, small molecule generation, and nucleic acid protein complex generation respectively. Our work is similar in that we focus on protein structure generation. We agree more tasks is desirable, but could be the focus of future work to develop extensions to generalize the approach to new tasks.
> > >
> > > # References
> > >
> > > [1] Watson, J. L., Juergens, D., Bennett, N. R., Trippe, B. L., Yim, J., Eisenach, H. E., ... & Baker, D. (2023). De novo design of protein structure and function with RFdiffusion. _Nature_, _620_(7976), 1089-1100.
> > >
> > > [2] Frank, C., Khoshouei, A., Fuβ, L., Schiwietz, D., Putz, D., Weber, L., ... & Dietz, H. (2024). Scalable protein design using optimization in a relaxed sequence space. _Science_, _386_(6720), 439-445.
> > >
> > > [3] Bennett, N. R., Coventry, B., Goreshnik, I., Huang, B., Allen, A., Vafeados, D., ... & Baker, D. (2023). Improving de novo protein binder design with deep learning. _Nature Communications_, _14_(1), 2625.
> > >
> > > [4] Zambaldi, V., La, D., Chu, A. E., Patani, H., Danson, A. E., Kwan, T. O., ... & Wang, J. (2024). De novo design of high-affinity protein binders with AlphaProteo. _arXiv preprint arXiv:2409.08022_.
> > >
> > > [5] Yim, J., Trippe, B.L., De Bortoli, V., Mathieu, E., Doucet, A., Barzilay, R. &amp; Jaakkola, T.. (2023). SE(3) diffusion model with application to protein backbone generation. <i>Proceedings of the 40th International Conference on Machine Learning</i>, in <i>Proceedings of Machine Learning Research</i> 202:40001-40039 Available from https://proceedings.mlr.press/v202/yim23a.html.
> > >
> > > [6] Lin, Y. &amp; Alquraishi, M.. (2023). Generating Novel, Designable, and Diverse Protein Structures by Equivariantly Diffusing Oriented Residue Clouds. <i>Proceedings of the 40th International Conference on Machine Learning</i>, in <i>Proceedings of Machine Learning Research</i> 202:20978-21002 Available from https://proceedings.mlr.press/v202/lin23a.html.
> > >
> > > [7] Avishek Joey Bose, Tara Akhound-Sadegh, Guillaume Huguet, Killian Fatras, Jarrid Rector-Brooks, Cheng-Hao Liu, Andrei Cristian Nica, Maksym Korablyov, Michael Bronstein, & Alexander Tong (2024). SE(3)-Stochastic Flow Matching for Protein Backbone Generation. In The International Conference on Learning Representations (ICLR).
> > >
> > > [8] https://www.nobelprize.org/prizes/chemistry/2024/press-release/
> > >
> > > [9] Jason Yim, Andrew Campbell, Emile Mathieu, Andrew Y. K. Foong, Michael Gastegger, Jose Jimenez-Luna, Sarah Lewis, Victor Garcia Satorras, Bastiaan S. Veeling, Frank Noe, Regina Barzilay, & Tommi Jaakkola (2024). Improved motif-scaffolding with SE(3) flow matching. Transactions on Machine Learning Research.
> > >
> > > [10] Ingraham, J., Garg, V., Barzilay, R., & Jaakkola, T. (2019). Generative Models for Graph-Based Protein Design. In _Advances in Neural Information Processing Systems_. Curran Associates, Inc..

---

> > > > ### Comment · Reviewer_S3eE · 2024-12-02
> > > > **Response**
> > > >
> > > > Thanks for the rebuttal. After reading the comments of other reviewers and considering the overall quality of this paper, I will keep the score as it is.

---

### Official Review · Reviewer_6bhQ · 2024-11-03

**Soundness:** 3
**Presentation:** 3
**Contribution:** 3
**Rating:** 5
**Confidence:** 5

**Summary:**

the paper proposes to split protein structural generations into 2 levels, namely latent (contact map) generation and structural (coordinates) generation. 2 other techs are proposed to enhance performance via gradient-guided conditional diffusion generation: long-range-contact (LRC) and predicted-aligned-error (PAE) guidances.

**Strengths:**

The motivation of combining 2-level generation is novel (at least in protein design) and promising. The framework allows one to connect semantics of proteins and their rigorous structures. For example, the guidance of PAE and LRC can be more elegantly incorporated through the new paradigm.

Section 5.2 is well-organized to show the benefits of LDM+SDM paradigm and the effect of PAE and LRC.

**Weaknesses:**

The main results in Table 2 indicates that existing structural diffusion methods are superior to the proposed one. I respect the authors for honestly presenting results that are less competitive, while the fact seems to be that latent diffusion does not work well on structural generation. I would reckon the authors to explore their methods on more specific applications to find better evidence for reviewers to support the submission,  eg language-conditioned protein design (to excavate the advantage of LDMs), loop design for antibodies or motif-scaffolding for enzymes.

**Questions:**

The authors may avoid calling their method *LSD*, as it causes negative associations.

several typos: L094, L285.

---

> ### Author Response · Authors · 2024-11-18
> **Response part 1**
>
> We thank reviewer 6bhQ for their time and review of our work. We appreciate the acknowledgement of the method’s promise and novelty. **We notice the review is relatively short with few concrete action items**. We believe we have addressed all weaknesses and questions. If the reviewer agrees we would very much appreciate a score increase. If not, we are happy to continue discussing any remaining weaknesses and questions. Below we address the reviewer’s comments and concerns.
>
> > while the fact seems to be that latent diffusion does not work well on structural generation
>
> We respectfully disagree with this statement. Our results show latent diffusion is competitive with state-of-the-art SDMs and improves in efficiency (GENIE2 taking 2.3 min while LSD taking 0.3 min to sample). We argue an approach does not need to beat state-of-the-art if it is novel enough and competitive. Structure diffusion models have been actively developed over the past 3 years [1] whereas latent diffusion models have not seen the same adoption. Yet, latent diffusion models are the de facto approach in image and video generation [2, 3]. This necessitates research to understand why and if there are benefits with latent diffusion for protein backbone generation.
>
> Note the only other open sourced (continuous) latent diffusion approach is LatentDiff upon which we improve considerably. Our contribution is to present a latent diffusion approach that will broaden the diversity of machine learning techniques to protein structure generation.
>
> > I would reckon the authors to explore their methods on more specific applications to find better evidence for reviewers to support the submission, eg language-conditioned protein design (to excavate the advantage of LDMs), loop design for antibodies or motif-scaffolding for enzymes.
>
> We believe these applications are a follow-up work. Unconditional generation is the first problem to address when developing a new generative model. For example, FrameFlow, FoldFlow, GENIE all started with unconditional backbone generation [4, 5, 6] before moving towards concrete applications [7, 8, 9]. We are planning to pursue all-atom generation for specific protein design tasks as a follow-up (as stated in our discussion). We aimed to focus this paper on method development; in particular, finding the right hyperparameters and architecture to work for latent diffusion. With our manuscript already at 20+ pages (including app.), more tasks would overload the content for a single conference submission.
>
> > The authors may avoid calling their method LSD, as it causes negative associations.
>
> We may change the name for camera ready but not for discussion to avoid further confusion.
>
> # References
>
> [1] Yim, Jason, et al. "Diffusion models in protein structure and docking." Wiley Interdisciplinary Reviews: Computational Molecular Science 14.2 (2024): e1711.
>
> [2] Rombach, Robin, et al. "High-resolution image synthesis with latent diffusion models." _Proceedings of the IEEE/CVF conference on computer vision and pattern recognition_. 2022.
>
> [3] Brooks, Tim, et al. "Video generation models as world simulators." 2024-03-03]. https://openai. com/research/video-generation-modelsas-world-simulators (2024).
>
> [4] Yim, Jason, et al. "Fast protein backbone generation with SE (3) flow matching." arXiv preprint arXiv:2310.05297 (2023).
>
> [5] Bose, Avishek Joey, et al. "Se (3)-stochastic flow matching for protein backbone generation." arXiv preprint arXiv:2310.02391 (2023).
>
> [6] Lin, Yeqing, and Mohammed AlQuraishi. "Generating novel, designable, and diverse protein structures by equivariantly diffusing oriented residue clouds." arXiv preprint arXiv:2301.12485 (2023).
>
> [7] Yim, Jason, et al. "Improved motif-scaffolding with se (3) flow matching." arXiv preprint arXiv:2401.04082 (2024).
>
> [8] Huguet, Guillaume, et al. "Sequence-Augmented SE (3)-Flow Matching For Conditional Protein Backbone Generation." arXiv preprint arXiv:2405.20313 (2024).
>
> [9] Lin, Yeqing, et al. "Out of Many, One: Designing and Scaffolding Proteins at the Scale of the Structural Universe with Genie 2." arXiv preprint arXiv:2405.15489 (2024).

---

> > ### Author Response · Authors · 2024-11-23
> >
> > Dear reviewer 6bhQ,
> >
> > As the discussion period is nearing its end, we ask the reviewer to respond if their concerns and questions have been addressed. We are happy to engage further and attempt to answer any more questions or concerns. If the concerns have been addressed, then we ask the reviewer to reconsider the rating.
> >
> > Thank you

---

### Official Review · Reviewer_Bnxy · 2024-11-07

**Soundness:** 3
**Presentation:** 3
**Contribution:** 3
**Rating:** 3
**Confidence:** 5

**Summary:**

Summary: This paper focuses on the task of protein backbone generation. The authors propose a new hierarchical method, called LSD, to generate protein backbones from coarse-grained to fine-grained. The first stage of LSD is to generate protein contact map (coarse-grained), and the second stage is to generate protein backbone conditioned on the contact map. An autoencoder (map between protein backbone, latent representation, and contact map) and two diffusion models are trained in this method. The model can generate diverse, designable, and novel protein backbones, and is comparable to current SOTA.

**Strengths:**

1. Although this paper is based on several previous papers and requires some background knowledge, it is still well-written and not difficult to understand.
2. The tables and figures are informative.
3. The results are comparable to current SOTA.

**Weaknesses:**

1. About the model design and corresponding ablation study: The current ablation study is more about hyperparameter search, missing support for the method design, for example, why do we need the contact map. One ablation is a pure latent diffusion model without contact map (similar to GeoLDM).
2. About the task: this paper only focuses on backbone generation, can it be extended to all-atom generation?
3. About baseline methods: AlphaFold3 also has a diffusion step. How to compare this method with AlphaFold3?
4. About the training: This method includes several training stages and is not trained end-to-end, which may limit the final performance. Can it be trained end-to-end? If so, what are the results?
5. In addition to the efficiency, what is the advantage over existing SDM (such as GENIE2)? We see that GENIE2 achieves good results. In addition, I am wondering why this method is more efficient than other methods. I think the diffusion step should be quite slow, and this method has two diffusion parts. In addition, from section 5.2, we know that k=4. And the input feature is 3*3=9. Therefore, the latent doesn’t compress the input a lot.

Other points:
1. For LDM, how does DiT deal with different input sizes? Do we need any other design?
2. Line 285: missing reference
3. Add reference to the contact map definition used here (line 158)

**Questions:**

See Weaknesses

---

> ### Author Response · Authors · 2024-11-18
> **Response part 1**
>
> We thank reviewer Bnxy for their time and review of our work. We appreciate the acknowledgement of the paper’s clarity and SOTA results with a novel model. Below we address the reviewer’s comments and concerns. We believe we have addressed all weaknesses and questions. If the reviewer agrees we would very much appreciate a score increase. If not, we are happy to continue discussing any remaining weaknesses and questions.
>
> > About the model design and corresponding ablation study: The current ablation study is more about hyperparameter search, missing support for the method design, for example, why do we need the contact map. One ablation is a pure latent diffusion model without contact map (similar to GeoLDM).
>
> In our initial submission, we performed a method design ablation in Table 6 of using DiT vs. DiT with RoPE vs. vanilla transformer as the latent diffusion architecture. As motivated in our introduction, the contact map is a coarse representation of protein structures that complements the structure diffusion in FrameFlow. Without a contact map, FrameFlow is unable to perform well on the AFDB dataset as shown in Table 2.
>
> We performed more ablations of the method design as requested. Using our best hyperparameter settings, we trained the autoencoder without contact maps where only the latents were provided into the FrameFlow architecture. In addition, we investigated the need for two stage autoencoder and FrameFlow training. Each set-up was trained for an equal number of epochs then evaluated in how well FrameFlow could reconstruct encoded structures.
>
> Specifically, for a structure $x$ in the validation set, we sampled latents $z \sim p_\phi(x)$ then sampled a structure using FrameFlow $\hat{x} \sim FrameFlow(z)$. We report the reconstruction RMSD as the average RMSD over the validation set $1/N \sum_{i=1}^N RMSD(x_i, \hat{x}_i)$ where $N$ is the validation set size. The results are in Table 7 and reproduced below where the reconstruction RMSD shows the two stage training with contact map loss leads to the lowest reconstruction RMSD. This shows both components are necessary for FrameFlow to learn how to decode the latents. We have included this discussion in Appendix C.1.
>
> | Training                                          | Contact map Loss | Reconstruction RMSD |
> | ------------------------------------------------- | ---------------- | ------------------- |
> | End-to-end training of Autoencoder and FrameFlow. | No               | 12.0                |
> |                                                   | Yes              | 4.1                 |
> | Two stage training of Autoencoder then FrameFlow  | Yes              | **1.8**                 |
>
> Lastly, the reviewer points out GeoLDM as another option. There are many ways to formulate latent diffusion and we emphasize that our approach is just one. Our approach is the first approach to be competitive with state-of-the-art SDMs.
>
> > About the task: this paper only focuses on backbone generation, can it be extended to all-atom generation?
>
> Yes! In line 534 of the diffusion, we explicitly mention extending to all-atom generation. This introduces new challenges such as how to deal with ligands and side chains that we are interested to explore as a follow-up.
>
> > About baseline methods: AlphaFold3 also has a diffusion step. How to compare this method with AlphaFold3?
>
> While AlphaFold3 is a diffusion model, it is not comparable since AlphaFold3 is a structure prediction method where a structure is generated conditioned on a sequence. Our approach is a sequence-agnostic structure generation method.
>
> > About the training: This method includes several training stages and is not trained end-to-end, which may limit the final performance. Can it be trained end-to-end? If so, what are the results?
>
> Our previous comment already addresses joint autoencoder and FrameFlow training which we found to be suboptimal. We mention end-to-end autoencoder and latent diffusion training in our initial submission, line 187, “While end-to-end training of all models is possible, we found this to be unstable and difficult to optimize. We instead use a training procedure inspired by Rombach et al. (2022) where the autoencoder is frozen during latent diffusion training.” Our end-to-end training results were very poor. We found near 0 designability and the loss to be unstable. As mentioned, this is consistent with state-of-the-art latent diffusion models in images that are trained in multiple stages rather than end-to-end [1].

---

> > ### Author Response · Authors · 2024-11-18
> > **Response part 2**
> >
> > > In addition to the efficiency, what is the advantage over existing SDM (such as GENIE2)? We see that GENIE2 achieves good results. In addition, I am wondering why this method is more efficient than other methods. I think the diffusion step should be quite slow, and this method has two diffusion parts.
> >
> > We describe the advantages over each SDM:
> > * GENIE2: Our method is far more efficient (eight times faster) and captures a more diverse secondary structure distribution as shown in Figure 7. Note GENIE2 collapses to sampling mostly alpha helices.
> > * FrameFlow AFDB: Our method far surpasses FrameFlow trained on AFDB. Note FrameFlow is part of LSD and so our intention is to show the inclusion of latent diffusion is complementary to scale FrameFlow to larger datasets.
> > * RFdiffusion and FrameFlow: With different guidances, we achieve superior metrics in each category. We demonstrate the aspect of controllability in our model to optimize different desirable properties.
> >
> > Our method is more efficient than GENIE2 since they utilize triangle multiplicative update layers which are O(L^3) where L is the protein length. One motivation for latent diffusion is we can utilize cheap and fast transformer implementations. While we use two diffusion steps, each diffusion only scales O(L^2) and is faster than GENIE2.
> >
> > > In addition, from section 5.2, we know that k=4. And the input feature is 3*3=9. Therefore, the latent doesn’t compress the input a lot.
> >
> > We don’t claim compression as being an advantage. The advantage is first modeling in an invariant feature space with scalable architectures, DiT, that scales well to AFDB. We have shown only structure diffusion with FrameFlow is unable to scale to larger datasets.
> >
> > > For LDM, how does DiT deal with different input sizes? Do we need any other design?
> >
> > DiT is a transformer which natively handles different sequence lengths. We refer to the original transformer paper if the reviewer would like to understand how transformers handle different input sizes. [2]
> >
> > > Line 285: missing reference
> >
> > Thank you! We have corrected this.
> >
> > > Add reference to the contact map definition used here (line 158)
> >
> > Thank you! We have added a justification and reference for the contact map definition, “8A is commonly used to define a contact [3]. ”
> >
> > # References
> >
> > [1] Rombach, Robin, et al. "High-resolution image synthesis with latent diffusion models." _Proceedings of the IEEE/CVF conference on computer vision and pattern recognition_. 2022.
> >
> > [2] Vaswani, A. "Attention is all you need." Advances in Neural Information Processing Systems (2017).
> >
> > [3] Hopf, Thomas A., et al. "Sequence co-evolution gives 3D contacts and structures of protein complexes." elife 3 (2014): e03430.

---

> > > ### Author Response · Authors · 2024-11-23
> > >
> > > Dear reviewer Bnxy,
> > >
> > > As the discussion period is nearing its end, we ask the reviewer to respond if their concerns and questions have been addressed.  We are happy to engage further and attempt to answer any more questions or concerns. If the concerns have been addressed, then we ask the reviewer to reconsider the rating.
> > >
> > > Thank you

---

> > > > ### Comment · Reviewer_Bnxy · 2024-11-27
> > > >
> > > > Thanks for the authors’ rebuttal. But some of my concerns are not well addressed, and hope the authors can provide more explanation.
> > > > 1. About my W5: thanks for providing advantages over each SDM and explaining why LSD is more efficient than GENIE2. I am wondering if LSD is also more efficient than FrameFlow and RFdiffusion.
> > > > 2. Based on my understanding, one of the motivations is that FrameFlow can’t scale to large datasets. Can you provide some intuitive explanation on why it can’t scale?
> > > > 3. About my Q1: The authors didn’t answer my question directly. Since the original input to DiT is image with fixed sequence length, I think it is worth providing more details on how to apply DiT here.
> > > >
> > > > Thanks.

---

### Official Review · Reviewer_NGTU · 2024-11-08

**Soundness:** 3
**Presentation:** 3
**Contribution:** 3
**Rating:** 6
**Confidence:** 4

**Summary:**

This paper proposed a hierarchical framework for protein backbone design containing two stages. The first stage is built upon a latent diffusion model: a structure-to-contact autoencoder based on ProteinMPNN encoder and contact predicter was pretrained to encode the protein structure to $L \times K$ representations. Then a modified Diffusion Transformer was trained in the latent space to generate synthetic representations. In the second stage, a modified version of FrameFlow was trained to design the atomic coordinates from the contact map. Additionally, two further guidance factors—long-range contact and predicted alignment error—were introduced to jointly enhance designability and diversity.

**Strengths:**

1) The overall architecture is well organized.
2) The clear mathematical  formulation helps to understand the real pipeline of the models.
3)  Evaluations across various aspects highlight the contributions of different factors.

**Weaknesses:**

1) The motivation is a bit unclear to me, as there are several approaches to get rid of equivariance issue or to develop SE(3)-equivariant models. For instance, FoldingDiff uses internal coordinates, and FrameFlow’s main architecture is SE(3)-equivariant. Intuitively, I don’t see a significant advantage to applying diffusion models on contact maps.
2) I'm unclear on certain aspects of the model design, such as why only node-wise features from ProteinMPNN are used for the latent representation, and why FrameFlow was chosen for recovering atomic coordinates, given that there are other models for structure prediction based on contact maps.
3) It would be better to apply evaluation metrics to natural samples in order to reveal any biases in the evaluation pipeline. Additional analysis on how the models perform across proteins of varying sizes, classes, or folds would also be beneficial. Besides secondary structures, other biological analysis such as torsional angles, clashes and Ramachandran plots are needed.
4) Some typos need correction, such as the citation of Chroma in Section 4.

**Questions:**

1) Why did you choose contact map rather than distance matrix while the latter is also SE(3)-invariant and more informative? There are also constraints on the contact map, such as neighboring residues in a sequence being in contact and certain patterns reflecting the corresponding secondary structure. Did you apply any methods to enforce these constraints?
2) While FrameFlow was developed based on continuous normalizing flows (CNF) models, why didn't you also apply CNF for the first stage?
3) Could you please explain what is the designable pairwise TM-score for and why it is the smaller the better here?

---

> ### Author Response · Authors · 2024-11-18
> **Response part 1**
>
> We thank reviewer NGTU for their time and review of our work. We appreciate the acknowledgment of the paper’s clarity and insightful evaluations. Below we address the reviewer’s comments and concerns. We believe we have addressed all weaknesses and questions. If the reviewer agrees we would very much appreciate a score increase. If not, we are happy to continue discussing any remaining weaknesses and questions.
>
> > The motivation is a bit unclear to me, as there are several approaches to get rid of equivariance issue or to develop SE(3)-equivariant models. For instance, FoldingDiff uses internal coordinates, and FrameFlow’s main architecture is SE(3)-equivariant. Intuitively, I don’t see a significant advantage to applying diffusion models on contact maps.
>
> To clarify the problem and motivation, we have rewritten the introduction with this as the first sentences.
>
> *“A challenge across diffusion models for protein backbone generation has been scaling to large datasets: ideally benefiting from improved diversity and generalization, but this empirically results in unwanted biases from low quality protein structures (Huguet et al., 2024). In this work, **we aim to develop a diffusion model that scales to the AlphaFold DataBase (AFDB) (Varadi et al., 2022) with the ability to control for desired properties**.”*
>
> Removing equivariance is not our goal but rather to improve FrameFlow performance with contact maps. We empirically demonstrate the scaling issue in section 5.2 where a FrameFlow model trained on the large AFDB dataset performs very poorly (Figure 2). We improve upon FrameFlow by surpassing its performance in Table 2 (i.e. 23% compared to 94% with LSD).
>
> FoldingDiff is another possible approach. However, based on their results in [1], we are unable to conclude that it can scale to large datasets. To reiterate, our approach is just one novel approach to overcome shortcomings in SE(3) equivariant models such as FrameFlow.
>
> Finally, latent diffusion has not been adequately explored in protein generation as much as it has in other domains such as computer vision [2]. We emphasize that our approach is the first latent diffusion model to be competitive with structure diffusion models.
>
> > I'm unclear on certain aspects of the model design, such as why only node-wise features from ProteinMPNN are used for the latent representation
>
> We aim to avoid the latents growing $O(L^2)$ where $L$ is the length of the protein. By only using the node features, we keep the latents $O(L)$. Furthermore, we found the ProteinMPNN autoencoder to work sufficiently well with 0.99 ROCAUC and 0.99 PRAUC in reconstructing contact maps when only the node-level latents were used.
>
> > And why FrameFlow was chosen for recovering atomic coordinates, given that there are other models for structure prediction based on contact maps.
>
> Our method is not for structure prediction. Note structure prediction takes in a sequence and predicts the folded structure.  Our method is for structure generation without a sequence. We chose to use FrameFlow since its architecture is essentially AlphaFold2’s structure module that naturally takes in the contact map as input. See FrameDiff’s architecture in Figure 2 [3] that FrameFlow uses. Other networks could be used as well as mentioned in line 188.
>
> > It would be better to apply evaluation metrics to natural samples in order to reveal any biases in the evaluation pipeline.
>
> Our evaluation procedure follows a long line of existing works, i.e. [3,4,5,6], many of which we include as baselines and are published at machine learning conferences. By natural samples, we assume the reviewer is referring to the training dataset, AFDB in our case. Prior works [6] have found AFDB is highly diverse but contains low designable protein structures. We quantified this in Table 9 where AFDB has 12% designability with high diversity, 65% of examples belong in a unique cluster. As stated in our motivation, we wish to benefit from the increased diversity while keeping high designability – as done in prior works.
>
> > Additional analysis on how the models perform across proteins of varying sizes, classes, or folds would also be beneficial.
>
> We are unsure what the reviewer is asking for since our method in its present form is structure generation model. It is not a prediction model that we can use to evaluate across different protein categories.
>
> We have added figure 10 showing designability across length. We see that generally longer lengths lead to worse designability which has been reported in other works [3, 4, 5, 6]. We acknowledge that adding a capability for fold or class conditioning is an interesting future direction for LSD.

---

> > ### Author Response · Authors · 2024-11-18
> > **Response part 2**
> >
> > > Besides secondary structures, other biological analysis such as torsional angles, clashes and Ramachandran plots are needed.
> >
> > We have added figure 11 in the appendix that shows a Ramachandran plot of the dihedral angles between samples from LSD and 1000 structures randomly sampled from the training set. We find the distributions are distributionally the same.
> >
> > We have also computed atom clashes by calculating all atomic pairwise distances and find none are clashing. A clash is defined to occur if the van der Waals radii of two atoms overlap which can be detected if the pairwise distance is <0.4 A.
> >
> > > Why did you choose contact map rather than distance matrix while the latter is also SE(3)-invariant and more informative?
> >
> > Using a distance matrix would defeat the purpose of our approach since a distance matrix fully specifies a protein structure and would not require a hierarchical model like LSD. We found contact maps to be sufficient as the coarse protein structure representation while structure diffusion with FrameFlow adds fine-grained details. ProteinSGM [7] is a score-based generative model over distance matrices and torsion angles. We initially did not include this model as a baseline since their results were far below other models. However, to address this comment, we have included this method as a baseline in our updated manuscript where we find LSD is able to outperform ProteinSGM on all metrics except DPT. It is worth noting DPT is not as commonly used as Diversity since Diversity is more interpretable as the number of unique designable proteins. This provides a sense of distance vs. contact map generation.
> >
> > | Method                                   | Des | Div | DPT  | Nov  | SDD  | H/S/C    |
> > | ---------------------------------------- | --- | --- | ---- | ---- | ---- | -------- |
> > | ProteinSGM: pairwise distance generation | 49% | 122 | **0.37** | 0.69 | 0.51 | 61/10/29 |
> > | LSD_J (ours): contact map generation     | **74%** | **296** | 0.53 | **0.66** | **0.26** | 57/15/28 |
> >
> > > There are also constraints on the contact map, such as neighboring residues in a sequence being in contact and certain patterns reflecting the corresponding secondary structure. Did you apply any methods to enforce these constraints?
> >
> > We did not apply any constraints since with deep learning the model should learn these constraints. We find that neighboring residue being in contact is always satisfied. Figure 6 shows that the generated contact maps are able to satisfy different secondary structures. We believe it is a strength that the model can learn constraints without explicitly enforcing them.
> >
> > The goal of LSD is to model coarse grained details first before proceeding to model fine-grained details with FrameFlow. Figure 3 attempts to answer how realistic the contact maps are by comparing the sampled contact maps with structures from FrameFlow. The close agreement indicates the contact maps are able to satisfy nearly all physical constraints.
> >
> > > While FrameFlow was developed based on continuous normalizing flows (CNF) models, why didn't you also apply CNF for the first stage?
> >
> > In our initial submission, we ablated the CNF equivalent of latent diffusion by removing noise and sampling with an ordinary differential equation (ODE). The results are provided in Table 8 where we see diffusion (with noise) works better. We acknowledge that latent diffusion has a plethora of hyperparameters to tune and experiment with [2]. We believe we have done a sufficient number of ablations and hyperparameter sweeps in section 5.2 and Appendix C for the scope of a ML conference paper.
> >
> > > Could you please explain what is the designable pairwise TM-score for and why it is the smaller the better here?
> >
> > Our metrics are explained in Appendix B.3. Designability pairwise TM-score is described on lines 957-959. We copy this verbatim, “Reporting the number of clusters can be biased since there are many hyperparameters and algorithms for clustering. To present an unbiased view of diverisy, we report the average TM-score of all the pairwise TM-scores between designable samples.” Of note, this metric has been reported in previous works [6, 8].

---

> > > ### Author Response · Authors · 2024-11-18
> > > **Response part 3**
> > >
> > > # References
> > >
> > > [1] Wu, Kevin E., et al. "Protein structure generation via folding diffusion." _Nature communications_ 15.1 (2024): 1059.
> > >
> > > [2] Rombach, Robin, et al. "High-resolution image synthesis with latent diffusion models." _Proceedings of the IEEE/CVF conference on computer vision and pattern recognition_. 2022.
> > >
> > > [3] Yim, Jason, et al. "SE (3) diffusion model with application to protein backbone generation." _arXiv preprint arXiv:2302.02277_(2023).
> > >
> > > [4] Yim, Jason, et al. "Improved motif-scaffolding with se (3) flow matching." _arXiv preprint arXiv:2401.04082_ (2024).
> > >
> > > [5] Watson, Joseph L., et al. "De novo design of protein structure and function with RFdiffusion." _Nature_ 620.7976 (2023): 1089-1100.
> > >
> > > [6] Huguet, Guillaume, et al. "Sequence-Augmented SE (3)-Flow Matching For Conditional Protein Backbone Generation." _arXiv preprint arXiv:2405.20313_ (2024).
> > >
> > > [7] Lee, Jin Sub, Jisun Kim, and Philip M. Kim. "Score-based generative modeling for de novo protein design." _Nature Computational Science_ 3.5 (2023): 382-392.
> > >
> > > [8] Bose, Avishek Joey, et al. "Se (3)-stochastic flow matching for protein backbone generation." _arXiv preprint arXiv:2310.02391_(2023).

---

> > > ### Comment · Reviewer_NGTU · 2024-11-22
> > > **Further questions about the model architecture.**
> > >
> > > I appreciate the authors' efforts in addressing my questions, which greatly helped me better understand the paper, particularly the motivation and design process. Inspired by Reviewer Bnxy’s comments, I have a few additional questions about the model architecture. Although Transformers can handle sequences of varying lengths, DiT was initially developed for images and utilizes patchification to create fixed-length sequences. In the code of DiT, padding masks were not applied. Could you provide more details on this—specifically, whether you introduced padding masks or treated paddings as special tokens? Additionally, since DiT was designed for conditional image generation, did you simply remove the condition inputs and the associated concatenation module? Thank you.

---

> > > > ### Author Response · Authors · 2024-11-22
> > > > **Model architecture**
> > > >
> > > > We are glad our response helped answer the reviewer's questions! Below we provide a direct and detailed answer to the architecture questions.
> > > >
> > > > **Direct answer the architecture questions**: We do not introduce padding tokens. Since our latents are variable length, we simply set latents to 0 if they are unused and are careful about masking during each neural network operation such that the model only sees unmasked latents. We remove condition inputs (other than the time embedding) and concatenation module since we do not do conditional generation (though this is a logical next step of our method).
> > > >
> > > > **More detailed answer**: As the reviewer states, DiT works by taking spatial latents of shape $I\times I \times C$ with $I$ as the spatial dimension and $C$ as the latent dimensionality then patchify into a sequence of length $T = (I/p)^2$ where $p$ is the patch dimension (figure 2 of [1]). Each image input is processed to have the same dimensionality and $p$ is a hyperparameter that is set once at the start of training. Therefore, $I, p, T$ are the same for each image input. Padding tokens and masking during attention are not needed.
> > > >
> > > > Unlike images, proteins have variable length and it is unclear how to process proteins such that they have the same length and number of latent. One of our baselines, LatentDiff [2], attempted to encode protein structures into a fixed number of latents. However, in our benchmark, we see LatentDiff encounters the problem of having to *sample* the protein length which leads it to always sampling very short protein lengths, nothing above length 100, even with 10,000 samples (see Table 2 of our manuscript). We observed the issues with LatentDiff early on and decided to use variable length latents. Specifically for a protein of length $L$ the latent shape is $L \times C$. To train on proteins of different lengths in a batch, we use masking throughout DiT such that the model never sees unused latents which are set to 0.
> > > >
> > > > Whether a fixed or variable length latent space is a very interesting question. Concurrent to our work, Lu et al. [3] explored if an adaptive truncation of $L$ is possible when evaluation of ESMFold on protein structure prediction. Their findings were promising though have yet to be explored for generation tasks. Our work provides one such exploration of using variable length latents for protein generation that we hope the research community can be inspired by.
> > > >
> > > > [1] Peebles, William, and Saining Xie. "Scalable diffusion models with transformers." Proceedings of the IEEE/CVF International Conference on Computer Vision. 2023.
> > > >
> > > > [2] Fu, Cong, et al. "A latent diffusion model for protein structure generation." Learning on Graphs Conference. PMLR, 2024.
> > > >
> > > > [3] Lu, Amy X., et al. "Tokenized and Continuous Embedding Compressions of Protein Sequence and Structure." _bioRxiv_(2024): 2024-08.

---

### Author Response · Authors · 2024-11-18
**Global response**

We thank all reviewers and the area chairs for their service in reviewing and providing helpful feedback. In the global response, we provide an overview of the main changes to our work as well as highlight common questions. First, we summarize the novelty and contributions of our work.
* **Contribution**: We present a novel latent diffusion model, LSD, for protein backbone generation that can scale to the AlphaFold DataBase (AFDB) with the ability to guide samples towards higher designability and diversity. LSD achieves competitive performance with state-of-the-art structure diffusion models (i.e. GENIE2) while being **eight times** faster to sample with.
* **Novelty**: LSD is the first latent diffusion model for contact map generation and first demonstration of classifier guidance for protein structure generation. We develop guidance for Predicted Alignment Error (PAE) and Long Range Contacts (LRC) guidance which can be extended for new guidance properties.

# Main changes to manuscript:

* **New introduction**. In response to reviewer NGTU, the introduction has been rewritten to clearly state our contribution. We quote the first two sentences here: “A challenge across diffusion models for protein backbone generation has been scaling to large datasets: ideally benefiting from improved diversity and generalization, but this empirically results in unwanted biases from low quality protein structures. In this work, we aim to develop a diffusion model that scales to the AlphaFold DataBase (AFDB) with the ability to control for desired properties.”
* **New ablations**. In response to reviewer Bnxy, Appendix C.1 now includes an ablation study to show the benefits of the contact map loss and multi-stage training. We use the reconstruction RMSD to quantify how well LSD can autoencoder the validation set.

| Training                                          | Contact map Loss | Reconstruction RMSD |
| ------------------------------------------------- | ---------------- | ------------------- |
| End-to-end training of Autoencoder and FrameFlow. | No               | 12.0                |
|                                                   | Yes              | 4.1                 |
| Two stage training of Autoencoder then FrameFlow  | Yes              | **1.8**                 |

* **New results**. Reviewer NGTU suggested analyzing the Ramachandran plots, steric clashes, and results as a function of length. We have added these analyses in Figure 8, 9 and Appendix C.1. We have added a new baseline, ProteinSGM [1], to directly compare our contact map generation to ProteinSGM’s pairwise distance generation method. We surpass ProteinSGM on all metrics except Designable Pairwise TM (DPT). It is worth noting DPT is not as commonly used as Diversity since Diversity is more interpretable as the number of unique designable proteins.

| Method                                   | Des | Div | DPT  | Nov  | SDD  | H/S/C    |
| ---------------------------------------- | --- | --- | ---- | ---- | ---- | -------- |
| ProteinSGM: pairwise distance generation | 49% | 122 | **0.37** | 0.69 | 0.51 | 61/10/29 |
| $\text{LSD}_\text{J}$ (ours): contact map generation     | **74%** | **296** | 0.53 | **0.66** | **0.26** | 57/15/28 |

# Common comments

* **Clarity in presentation and mathematical details**. Reviewers NGTU, Bnxy, 6bhQ, and S3eE all commented on good paper quality and how the method was easy to follow and understand.
* **Comprehensive evaluations**. Reviewers NGTU, Bnxy, and 6bhQ all state how the evaluations provide insight into different factors of the method. Our updated results and ablations further strengthen our evaluation.
* **Competitive state-of-the-art results**. Reviewer Bnxy and S3eE noted the competitive results as a strength while 6bhQ stated the results are less competitive. Our results in section 5.4 are indeed competitive with state-of-the-art which is GENIE2 but achieves it with a **eight times faster model** due to not relying on expensive O(L^3) neural network architecture where L is the protein length. Furthermore, our approach is the first latent diffusion based approach to achieve state-of-the-art status since all previous approaches used diffusion models over atomic coordinates and frames.
* **Application to other tasks**. Reviewer Bnxy asks if the method can be extended to all-atom generation while reviewer 6bhQ asks for extension to loop design, motif-scaffolding, language conditioning. We appreciate these suggestions as they are all exciting future directions. However, we emphasize that the simplest task for a new generative model is unconditional generation on protein monomers. Now that we have successfully achieved strong unconditional generation performance, it is natural to move on to real protein design tasks.

# References

[1] Lee, Jin Sub, Jisun Kim, and Philip M. Kim. "Score-based generative modeling for de novo protein design." Nature Computational Science 3.5 (2023): 382-392.

---

### Meta-Review · Area_Chair_3feu · 2024-12-22

**Metareview:**

The paper does not receive positive support from the reviewers, who raised major concerns on this paper, and these concerns are not resolved during rebuttals.

**Additional Comments On Reviewer Discussion:**

Extensive discussions have been involved, but the reviewers are not convinced eventually.

---

### Decision · Program_Chairs · 2025-01-22

Reject